# A Framework for Bidirectional Decoding: Case Study in Morphological Inflection

**Marc E. Canby and Julia Hockenmaier**
University of Illinois at Urbana-Champaign
{marcec2,juliahmr}@illinois.edu

## Abstract

Transformer-based encoder-decoder models that generate outputs in a left-to-right fashion have become standard for sequence-to-sequence tasks. In this paper, we propose a framework for decoding that produces sequences from the "outside-in": at each step, the model chooses to generate a token on the left, on the right, or join the left and right sequences. We argue that this is more principled than prior bidirectional decoders. Our proposal supports a variety of model architectures and includes several training methods, such as a dynamic programming algorithm that marginalizes out the latent ordering variable. Our model sets state-of-the-art (SOTA) on the 2022 and 2023 shared tasks, beating the next best systems by over 4.7 and 2.7 points in average accuracy respectively. The model performs particularly well on long sequences, can implicitly learn the split point of words composed of stem and affix, and performs better relative to the baseline on datasets that have fewer unique lemmas (but more examples per lemma).[1]

## 1 Introduction

Transformer-based encoder-decoder architectures (Bahdanau et al., 2014; Vaswani et al., 2017) that decode sequences from left to right have become dominant for sequence-to-sequence tasks. While this approach is quite straightforward and intuitive, some research has shown that models suffer from this arbitrary constraint. For example, models that decode left-to-right are often more likely to miss tokens near the end of the sequence, while right-to-left models are more prone to making mistakes near the beginning (Zhang et al., 2019; Zhou et al., 2019a). This is a result of the "snowballing" effect, whereby the model's use of its own incorrect predictions can lead future predictions to be incorrect (Bengio et al., 2015; Liu et al., 2016).

We explore this issue for the task of morphological inflection, where the goal is to learn a mapping from a word's lexeme (e.g. the lemma walk) to a particular form (e.g. walked) specified by a set of morphosyntactic tags (e.g. V;V.PTCP;PST). This has been the focus of recent shared tasks (Cotterell et al., 2016, 2017, 2018; McCarthy et al., 2019; Vylomova et al., 2020; Pimentel et al., 2021; Kodner et al., 2022; Goldman et al., 2023). Most approaches use neural encoder-decoder architectures, e.g recurrent neural networks (RNNs) (Aharoni and Goldberg, 2017; Wu and Cotterell, 2019) or transformers (Wu et al., 2021).[2] To our knowledge, Canby et al. (2020) is the only model that uses bidirectional decoding for inflection; it decodes the sequence in both directions simultaneously and returns the one with higher probability.

In this paper, we propose a novel framework for bidirectional decoding that supports a variety of model architectures. Unlike previous work (§2), at each step the model chooses to generate a token on the left, generate a token on the right, or join the left and right sequences.

This proposal is appealing for several reasons. As a general framework, this approach supports a wide variety of model architectures that may be task-specific. Further, it generalizes L2R and R2L decoders, as the model can choose to generate sequences in a purely unidirectional fashion. Finally, the model is able to decide which generation order is best for each sequence, and can even produce parts of a sequence from each direction. This is particularly appropriate for a task like inflection, where many words are naturally split into stem and affix. For example, when producing the form walked, the model may chose to generate the stem

---

[1] Our code is available at https://github.com/marccanby/bidi_decoding/tree/main.

[2] Orthogonal to the concerns in this paper, various data augmentation schemes such as heuristic alignment or rule-based methods (Kann and Schütze, 2017; Anastasopoulos and Neubig, 2019) or the use of multilingual data (Bergmanis et al., 2017; McCarthy et al., 2019) have been proposed to improve these standard architectures.

walk from the left and the suffix ed from the right.

We explore several methods for training models under this framework, and find that they are highly effective on the 2023 SIGMORPHON shared task on inflection (Goldman et al., 2023). Our method improves by over 4 points in average accuracy over a typical L2R model, and one of our loss functions is particularly adept at learning split points for words with a clear affix. We also set SOTA on both the 2022 and 2023 shared tasks (Kodner et al., 2022), which have very different data distributions.

## 2 Prior Bidirectional Decoders

Various bidirectional decoding approaches have been proposed for tasks such as machine translation and abstractive summarization, including ones that use some form of regularization to encourage the outputs from both directions to agree (Liu et al., 2016; Zhang et al., 2019; Shan et al., 2019), or algorithms where the model first decodes the entire sequence in the R2L direction and then conditions on that sequence when decoding in the L2R direction (Zhang et al., 2018; Al-Sabahi et al., 2018). Still more methods utilize synchronous decoding, where the model decodes both directions at the same time and either meet in the center (Zhou et al., 2019b; Imamura and Sumita, 2020) or proceed until each direction's hypothesis is complete (Zhou et al., 2019a; Xu and Yvon, 2021). Lawrence et al. (2019) allows the model to look into the future by filling placeholder tokens at each timestep.

## 3 A Bidirectional Decoding Framework

The following sections present a general framework for training and decoding models with bidirectional decoding that is irrespective of model architecture, subject to the constraints discussed in §3.3.

### 3.1 Probability Factorization

For unidirectional models, the probability of an L2R sequence $\overrightarrow{\boldsymbol{y}} = y_1 \cdots y_n$ or an R2L sequence $\overleftarrow{\boldsymbol{y}} = y_n \cdots y_1$ given an input $\boldsymbol{x}$ is defined as

$$P(\overrightarrow{\boldsymbol{y}}|\boldsymbol{x}) = \prod_{i=1}^{|\boldsymbol{y}|} P(\overrightarrow{y}_i | \overrightarrow{\boldsymbol{y}}_{<i}, \boldsymbol{x}) \quad (1)$$

$$P(\overleftarrow{\boldsymbol{y}}|\boldsymbol{x}) = \prod_{j=1}^{|\boldsymbol{y}|} P(\overleftarrow{y}_j | \overleftarrow{\boldsymbol{y}}_{<j}, \boldsymbol{x}) \quad (2)$$

where $\overrightarrow{y}_i = y_i$ or $\overleftarrow{y}_j = y_{n-j+1}$ is the $i$th or $j$th token in a particular direction. Generation begins with a start-of-sentence token; at each step a token is chosen based on those preceding, and the process halts once an end-of-sentence token is predicted.

In contrast, our bidirectional scheme starts with an empty prefix $ and suffix #. At each timestep, the model chooses to generate the next token of either the prefix or the suffix, and then whether or not to join the prefix and suffix. If a join is predicted, then generation is complete.

We define an *ordering* $\boldsymbol{o} = o^{(1)} \cdots o^{(n)}$ as a sequence of *left* and *right* decisions: that is, $o^{(t)} \in \{L, R\}$. We use $y^{(t)}$ to refer to the token generated at time $t$ under a particular ordering, and $\overrightarrow{\boldsymbol{y}}^{(\leq t)}$ and $\overleftarrow{\boldsymbol{y}}^{(\leq t)}$ to refer to the prefix and suffix generated up to (and including) time $t$.[3] An example derivation of the word walked is shown below:

| $o^{(t)}$ | $y^{(t)}$ | $\overrightarrow{\boldsymbol{y}}^{(\leq t)}$ | $\overleftarrow{\boldsymbol{y}}^{(\leq t)}$ |
|---|---|---|---|
| | | $ | # |
| $o^{(1)} = L$ | $y^{(1)} = w$ | $w | # |
| $o^{(2)} = L$ | $y^{(2)} = a$ | $wa | # |
| $o^{(3)} = R$ | $y^{(3)} = d$ | $wa | d# |
| $o^{(4)} = L$ | $y^{(4)} = l$ | $wal | d# |
| $o^{(5)} = R$ | $y^{(5)} = e$ | $wal | ed# |
| $o^{(6)} = L$ | $y^{(6)} = k$ | $walk | ed# |

Dropping the dependence on $\boldsymbol{x}$ for notational convenience, we define the joint probability of output sequence $\boldsymbol{y}$ and ordering $\boldsymbol{o}$ as

$$P(\boldsymbol{y}, \boldsymbol{o}) = \prod_{t=1}^{|\boldsymbol{y}|} P(o^{(t)} | \overrightarrow{\boldsymbol{y}}^{(<t)}, \overleftarrow{\boldsymbol{y}}^{(<t)}) \cdot$$
$$P(y^{(t)} | o^{(t)}, \overrightarrow{\boldsymbol{y}}^{(<t)}, \overleftarrow{\boldsymbol{y}}^{(<t)}) \cdot Q^{(t)} \quad (3)$$

where $Q^{(t)}$ is the probability of joining (or not joining) the prefix and suffix:

$$Q^{(t)} = \begin{cases} P(join \mid \overrightarrow{\boldsymbol{y}}^{(\leq t)}, \overleftarrow{\boldsymbol{y}}^{(\leq t)}) & \text{if } t = |\boldsymbol{y}| \\ 1 - P(join \mid \overrightarrow{\boldsymbol{y}}^{(\leq t)}, \overleftarrow{\boldsymbol{y}}^{(\leq t)}) & \text{otherwise} \end{cases}$$

### 3.2 Likelihood and MAP Inference

To compute the likelihood of a particular sequence $\boldsymbol{y}$, we need to marginalize over all orderings: $P(\boldsymbol{y}|\boldsymbol{x}) = \sum_{\boldsymbol{o}} P(\boldsymbol{y}, \boldsymbol{o}|\boldsymbol{x})$. Since we cannot enumerate all $2^{|\boldsymbol{y}|}$ orderings, we have developed an exact $O(|\boldsymbol{y}|^2)$ dynamic programming algorithm, reminiscent of the forward algorithm for HMMs.

To simplify notation, let $P_L(\overrightarrow{y}_i \mid \overrightarrow{\boldsymbol{y}}_{<i}, \overleftarrow{\boldsymbol{y}}_{<j})$ (or $P_R(\overleftarrow{y}_j \mid \overrightarrow{\boldsymbol{y}}_{<i}, \overleftarrow{\boldsymbol{y}}_{<j})$) be the probability of

---

[3]We use superscripts to refer to timesteps, and subscripts for sequence positions. Note that if, at a particular timestep $t$, we have prefix $\overrightarrow{\boldsymbol{y}}_{\leq i}$ and suffix $\overleftarrow{\boldsymbol{y}}_{\leq j}$, then $i + j = t$.

generating the $i$th token from the left (or the $j$th token from the right), conditioned on $\overrightarrow{\boldsymbol{y}}_{<i}$ and $\overleftarrow{\boldsymbol{y}}_{<j}$, the prefix and suffix generated thus far:

$$P_L(\overrightarrow{y}_i|\overrightarrow{\boldsymbol{y}}_{<i},\overleftarrow{\boldsymbol{y}}_{<j}) = P(L\,|\,\overrightarrow{\boldsymbol{y}}_{<i},\overleftarrow{\boldsymbol{y}}_{<j})\cdot P(\overrightarrow{y}_i|\,L,\overrightarrow{\boldsymbol{y}}_{<i},\overleftarrow{\boldsymbol{y}}_{<j})$$
$$P_R(\overleftarrow{y}_j|\overrightarrow{\boldsymbol{y}}_{<i},\overleftarrow{\boldsymbol{y}}_{<j}) = P(R\,|\,\overrightarrow{\boldsymbol{y}}_{<i},\overleftarrow{\boldsymbol{y}}_{<j})\cdot P(\overleftarrow{y}_j|R,\overrightarrow{\boldsymbol{y}}_{<i},\overleftarrow{\boldsymbol{y}}_{<j})$$

Let $Q_{ij}$ be the join probability for $\overrightarrow{\boldsymbol{y}}_{\leq i}$ and $\overleftarrow{\boldsymbol{y}}_{\leq j}$:

$$Q_{ij} = \begin{cases} P(join \mid \overrightarrow{\boldsymbol{y}}_{\leq i},\overleftarrow{\boldsymbol{y}}_{\leq j}) & \text{if } i+j=|\boldsymbol{y}| \\ 1 - P(join \mid \overrightarrow{\boldsymbol{y}}_{\leq i},\overleftarrow{\boldsymbol{y}}_{\leq j}) & \text{otherwise} \end{cases} \quad (4)$$

Finally, denote the joint probability of a prefix $\overrightarrow{\boldsymbol{y}}_{\leq i}$ and suffix $\overleftarrow{\boldsymbol{y}}_{\leq j}$ by $f[i,j]$.

We set the probability of an empty prefix and suffix (the base case) to 1:

$$f[0,0] = 1$$

The probability of a non-empty prefix $\overrightarrow{\boldsymbol{y}}_{\leq i}$ and empty suffix $\epsilon$ can be computed by multiplying $f[i-1,0]$ (the probability of prefix $\overrightarrow{\boldsymbol{y}}_{<i}$ and empty suffix $\epsilon$) by $P_L(\overrightarrow{y}_i \mid \overrightarrow{\boldsymbol{y}}_{<i},\epsilon)$ (the probability of generating $\overrightarrow{y}_i$) and the *join* probability $Q_{i0}$:

$$f[i,0] = f[i-1,0]\cdot P_L(\overrightarrow{y}_i|\overrightarrow{\boldsymbol{y}}_{<i},\epsilon)\cdot Q_{i0}$$

Analogously, we define

$$f[0,j] = f[0,j-1]\cdot P_R(\overleftarrow{y}_j|\epsilon,\overleftarrow{\boldsymbol{y}}_{<j})\cdot Q_{0j}$$

Finally, $f[i,j]$ represents the case where both prefix $\overrightarrow{\boldsymbol{y}}_{\leq i}$ and suffix $\overleftarrow{\boldsymbol{y}}_{\leq j}$ are non-empty. This prefix-suffix pair can be produced either by appending $\overrightarrow{y}_i$ to the prefix $\overrightarrow{\boldsymbol{y}}_{<i}$ and leaving the suffix unchanged, or by appending $\overleftarrow{y}_j$ to the suffix $\overleftarrow{\boldsymbol{y}}_{<j}$ and leaving the prefix unchanged. The sum of the probabilities of these cases gives the recurrence:

$$f[i,j] = f[i-1,j]\cdot P_L(\overrightarrow{y}_i|\overrightarrow{\boldsymbol{y}}_{<i},\overleftarrow{\boldsymbol{y}}_{\leq j})\cdot Q_{ij}+$$
$$f[i,j-1]\cdot P_R(\overleftarrow{y}_j|\overrightarrow{\boldsymbol{y}}_{\leq i},\overleftarrow{\boldsymbol{y}}_{<j})\cdot Q_{ij}$$

After filling out the dynamic programming table $f$, the marginal probability $P(\boldsymbol{y})$ can be computed by summing all entries $f[i,j]$ where $i+j=|\boldsymbol{y}|$:

$$P(\boldsymbol{y}) = \sum_{i,j} \mathbb{I}(i+j=|\boldsymbol{y}|)\cdot f[i,j]$$

If all local probabilities can be calculated in constant time, the runtime of this algorithm is $O(|\boldsymbol{y}|^2)$.

As an aside, the MAP probability, or the probability of the best ordering for a given sequence, can be calculated by replacing each sum with a max:

$$f[i,j] = \max\big(f[i-1,j]\cdot P_L(\overrightarrow{y}_i|\overrightarrow{\boldsymbol{y}}_{<i},\overleftarrow{\boldsymbol{y}}_{\leq j})\cdot Q_{ij},$$
$$f[i,j-1]\cdot P_R(\overleftarrow{y}_j|\overrightarrow{\boldsymbol{y}}_{\leq i},\overleftarrow{\boldsymbol{y}}_{<j})\cdot Q_{ij}\big)$$
$$\max_{\boldsymbol{o}} P(\boldsymbol{y},\boldsymbol{o}) = \max_{i,j}\big(\mathbb{I}(i+j=|\boldsymbol{y}|)\cdot f[i,j]\big)$$

The best ordering itself can be found with a backtracking procedure similar to Viterbi for HMM's.

## 3.3 Why does dynamic programming work?

Dynamic programming (DP) only works for this problem if the local probabilities (i.e. the token, join, and order probabilities) used to compute $f[i,j]$ depend *only* on the prefix and suffix corresponding to that cell, but *not* on a particular ordering that produced the prefix and suffix. This is similar to the how the Viterbi algorithm relies on the fact that HMM emission probabilities depend only on the hidden state and not on the path taken.

To satisfy this requirement, the model's architecture should be chosen carefully. Any model that simply takes a prefix and suffix as input and returns the corresponding local probabilities is sufficient. However, one must be careful if designing a model where the hidden representation is shared or reused across timesteps. This is particularly problematic if hidden states computed from *both* the prefix and suffix are reused. In this case, the internal representations will differ depending on the order in which the prefix and suffix were generated, which would cause a DP cell to rely on all possible paths to that cell − thus breaking the polynomial nature of DP.

## 3.4 Training

We propose two different loss functions to train a bidirectional model. Based on our probability factorization, we must learn the token, join, and order probabilities at each timestep.

Our first loss function $\mathcal{L}_{xH}(\theta)$ trains each of these probabilities separately using cross-entropy loss. However, since ordering is a latent variable, it cannot be trained with explicit supervision. Hence, we fix the order probability to be $0.5$ at each timestep, making all orderings equi-probable.

We then define $\mathcal{S}$ to contain the indices of all valid prefix-suffix pairs in a given sequence $\boldsymbol{y}$:

$$\mathcal{S} = \{(i,j)\mid 1\leq i,j,\leq|\boldsymbol{y}|;i+j\leq|\boldsymbol{y}|\}$$

Hence, $\mathcal{S}$ has $O(|\boldsymbol{y}|^2)$ elements.

Finally, we define a simple loss $\mathcal{L}_{xH}(\theta)$ that averages the cross-entropy loss for the token probabilities (based on the next token in $\overrightarrow{\boldsymbol{y}}$ or $\overleftarrow{\boldsymbol{y}}$) and join probabilities (based on whether the given prefix and suffix complete $\boldsymbol{y}$):

$$\mathcal{L}_{xH}(\theta) = \frac{1}{3}\Big(\overrightarrow{\mathcal{L}}(\theta) + \overleftarrow{\mathcal{L}}(\theta) + \mathcal{L}^{(join)}(\theta)\Big)$$
$$\overrightarrow{\mathcal{L}}(\theta) = -\frac{1}{|\mathcal{S}|}\sum_{(i,j)\in\mathcal{S}}\log P(\overrightarrow{y}_i \mid \overrightarrow{\boldsymbol{y}}_{<i},\overleftarrow{\boldsymbol{y}}_{<j},\boldsymbol{x};\theta)$$
$$\overleftarrow{\mathcal{L}}(\theta) = -\frac{1}{|\mathcal{S}|}\sum_{(i,j)\in\mathcal{S}}\log P(\overleftarrow{y}_j \mid \overrightarrow{\boldsymbol{y}}_{<i},\overleftarrow{\boldsymbol{y}}_{<j},\boldsymbol{x};\theta)$$

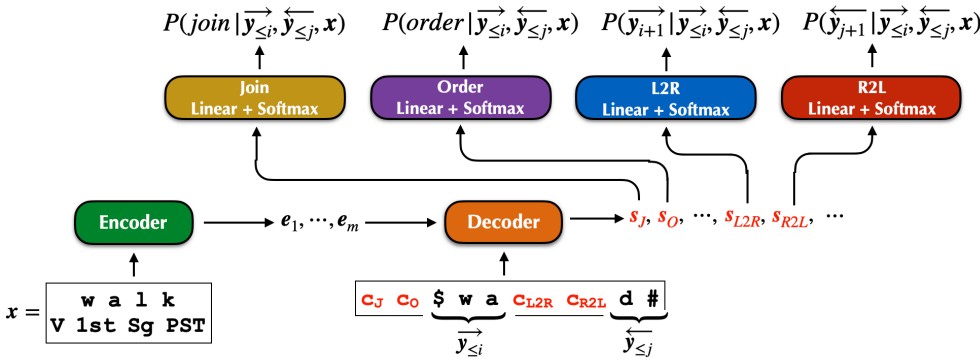

Figure 1: **Architecture for bidirectional decoding model.** Depicts the token inputs for the verb walked at timestep $t = 3$ with $\overrightarrow{y}_{\leq 2} = \$wa$ and $\overleftarrow{y}_{\leq 1} = d\#$. All inputs are surrounded by a rectangle.

$$\mathcal{L}^{(join)}(\theta) = -\frac{1}{|\mathcal{S}|} \sum_{(i,j) \in \mathcal{S}} \log Q_{ij}$$

where $Q_{ij}$ is defined as in Equation 4.

Due to the size of $\mathcal{S}$, this loss takes $O(|\boldsymbol{y}|^2)$ time to train.[4] Given that a typical unidirectional model takes $O(|\boldsymbol{y}|)$ time to train, we also propose an $O(|\boldsymbol{y}|)$ approach that involves sampling from $\mathcal{S}$; this is presented in Appendix F.

An alternative is to train with Maximum Marginal Likelihood (MML) (Guu et al., 2017; Min et al., 2019), which learns the order probabilities via marginalization. This is more principled because it directly optimizes $P(\boldsymbol{y} \mid \boldsymbol{x})$, the quantity of interest. The loss is given by $\mathcal{L}_{MML}(\theta) = -\log P(\boldsymbol{y}|\boldsymbol{x}; \theta)$, which is calculated with the dynamic programming algorithm described in §3.2.[5] Learning the order probabilities enables the model to assign higher probability mass to orderings it prefers and ignore paths it finds unhelpful.

This loss also requires $O(|\boldsymbol{y}|^2)$ time to train.

### 3.5 Decoding

The goal of decoding is to find $\boldsymbol{y}$ such that $\boldsymbol{y} = \text{argmax}_{\boldsymbol{y}} P(\boldsymbol{y}|\boldsymbol{x})$. Unfortunately, it is not computationally feasible to use the likelihood algorithm in §3.2 to find the *best* sequence $\boldsymbol{y}$, even with a heuristic like beam search. Instead, we use beam search to heuristically identify the sequence $\boldsymbol{y}$ and ordering $\boldsymbol{o}$ that maximize the *joint* probability $P(\boldsymbol{y}, \boldsymbol{o}|\boldsymbol{x})$:

$$\boldsymbol{y}, \boldsymbol{o} = \text{argmax}_{\boldsymbol{y}, \boldsymbol{o}} P(\boldsymbol{y}, \boldsymbol{o}|\boldsymbol{x})$$

---

[4]This assumes that local probabilities take $O(1)$ time to compute, which is not the case for most neural architectures; however, this section is about the runtime of the training algorithms without regard to model architecture.

[5]Appendix C describes how to train this loss in practice.

The formula for $P(\boldsymbol{y}, \boldsymbol{o}|\boldsymbol{x})$ is given by Equation 3.

Each hypothesis is a prefix-suffix pair. We start with a single hypothesis: an empty prefix and suffix, represented by start- and end-of-sentence tokens. At a given timestep, each hypothesis is expanded by considering the distribution over possible actions: adding a token on the left, adding a token on the right, or joining. The $k$ best continuations are kept based on their (joint) probabilities. Generation stops once all hypotheses are complete (i.e. the prefix and suffix are joined).

## 4   Model Architecture

Our architecture (Figure 1) is based on the character-level transformer (Wu et al., 2021), which has proven useful for morphological inflection. First, the input sequence $\boldsymbol{x}$ is encoded with a typical Transformer encoder; for the inflection task, this consists of the lemma (tokenized by character) concatenated with a separator token and set of tags.

Given a prefix $\overrightarrow{y}_{\leq i}$ and suffix $\overleftarrow{y}_{\leq j}$ (as well as the encoder output), the decoder must produce each direction's token probabilities, the join probability, and the order probability. We construct the input to the decoder by concatenating the prefix and suffix tokens with some special classification tokens:

$$\langle c_J, c_O, \overrightarrow{y}_1, ..., \overrightarrow{y}_i, c_{L2R}, c_{R2L}, \overleftarrow{y}_j, ..., \overleftarrow{y}_1 \rangle$$

The tokens $c_J$, $c_O$, $c_{L2R}$, and $c_{R2L}$ are special classification tokens that serve a purpose similar to the CLS embedding in BERT (Devlin et al., 2019). We feed this input to a Transformer decoder as follows:

$$\boldsymbol{s}_J, \boldsymbol{s}_O, ..., \boldsymbol{s}_{L2R}, \boldsymbol{s}_{R2L}, ... = \text{Decoder}(\langle \cdots \rangle)$$

These vectors are fed through their own linear layers and softmax, giving the desired probabilities:

$$P(join \mid \overrightarrow{y}_{\leq i}, \overleftarrow{y}_{\leq j}) = \text{Softmax}(\boldsymbol{s}_O V)$$

| | **Avg** | **# Langs** $\geq$ BL2 | **# Langs** ($p \leq 0.05$) $>$ BL2 | $=$ BL2 | $<$ BL2 |
|---|---|---|---|---|---|
| **L2R** | 80.26 | – | – | – | – |
| **R2L** | 79.65 | – | – | – | – |
| **BL2** | 82.59 | – | – | – | – |
| **xH** | 84.25 | 19/27 | **12/27** | 12/27 | 3/27 |
| **MML** | 81.43 | 9/27 | 5/27 | 11/27 | 11/27 |
| **xH-Rerank** | **84.38** | 18/27 | **12/27** | 13/27 | 2/27 |
| **MML-Rerank** | 81.50 | 9/27 | 5/27 | 12/27 | 10/27 |
| **BL2-xH** | 84.00 | **24/27** | **12/27** | 15/27 | **0/27** |
| **BL2-MML** | 83.54 | 18/27 | 7/27 | 17/27 | 3/27 |
| Goldman et al. (2023) | 81.6 | – | – | – | – |

Table 1: **Accuracies of Methods.** Accuracy averaged over all languages in the SIGMORPHON 2023 shared task, and number of languages whose accuracy equals or exceeds ($\geq$) the best baseline BL2. The entry Goldman et al. (2023) shows the accuracy of the next best system submitted to the shared task. Also shows number of languages with a *statistically significant* improvement ($>$) or degradation ($<$), or no statistically significant change ($=$), in accuracy compared with BL2 using a paired-permutation test (Zmigrod et al., 2022) with $\alpha = 0.05$. The best entry in each column is **bold**. See Table 9 in Appendix D for results by language.

$$P(order \mid \overrightarrow{\boldsymbol{y}}_{\leq i}, \overleftarrow{\boldsymbol{y}}_{\leq j}) = \text{Softmax}(\boldsymbol{s}_J U)$$
$$P(\overrightarrow{y}_i \mid \overrightarrow{\boldsymbol{y}}_{\leq i}, \overleftarrow{\boldsymbol{y}}_{\leq j}) = \text{Softmax}(\boldsymbol{s}_J \overrightarrow{W})$$
$$P(\overleftarrow{y}_j \mid \overrightarrow{\boldsymbol{y}}_{\leq i}, \overleftarrow{\boldsymbol{y}}_{\leq j}) = \text{Softmax}(\boldsymbol{s}_J \overleftarrow{W})$$

Since this architecture *does* have cross-attention between the prefix and suffix, the decoder hidden states for each prefix-suffix pair must be recomputed at each timestep to allow for DP (see §3.3).

## 5 Experimental Setup

**Datasets.** We experiment with inflection datasets for all 27 languages (spanning 9 families) from the SIGMORPHON 2023 shared task (Goldman et al., 2023). Each language has 10,000 training and 1,000 validation and test examples, and no lemma occurs in more than one of these partitions. We also show results on the 20 "large" languages from the SIGMORPHON 2022 shared task (Kodner et al., 2022), which has a very different sampling of examples in the train and test sets. A list of all languages can be found in Appendix A.

**Tokenization.** Both the lemma and output form are split by character; the tags are split by semicolon. For the 2023 shared task, where the tags are "layered" (Guriel et al., 2022), we also treat each open and closed parenthesis as a token. Appendix B describes the treatment of unknown characters.

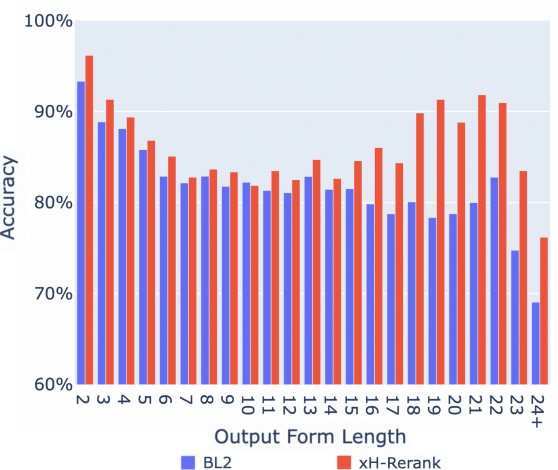

Figure 2: **Accuracies of xH-Rerank and BL2 by Word Length.** Average accuracies of BL2 and xH-Rerank models over all languages, grouped by length (number of characters) of the output form.

**Model hyperparameters.** Our models are implemented in fairseq (Ott et al., 2019). We experiment with small, medium, and large model sizes (ranging from ~240k to ~7.3M parameters). For each language, we select a model size based on the L2R and R2L unidirectional accuracies; this procedure is detailed in Appendix A.

The only additional parameters in our bidirectional model come from the embeddings for the 4 classification tokens (described in §4); hence, our unidirectional and bidirectional models have roughly the same number of parameters.

**Training.** We use a batch size of 800, an Adam optimizer ($\beta_1 = 0.9$, $\beta_2 = 0.98$), dropout of 0.3, and an inverse square root scheduler with initial learning rate $1e-07$. Training is halted if validation accuracy does not improve for 7,500 steps. All validation accuracies are reported in Appependix A.

**Inference.** Decoding maximizes joint probability $P(\boldsymbol{y}, \boldsymbol{o} \mid \boldsymbol{x})$ using the beam search algorithm of §3.5 with width 5. In some experiments, we rerank the 5 best candidates according to their marginal probability $P(\boldsymbol{y} \mid \boldsymbol{x})$, which can be calculated with dynamic programming (§3.2).

**Models.** We experiment with the following models (see Appendices D and F for more variants):

- **L2R & R2L**: Standard unidirectional transformer baselines, trained with the loss given in Equations 1 and 2.

- **BL2:** A naive "bidirectional" baseline that returns either the best L2R or R2L hypothesis based on which has a higher probability.

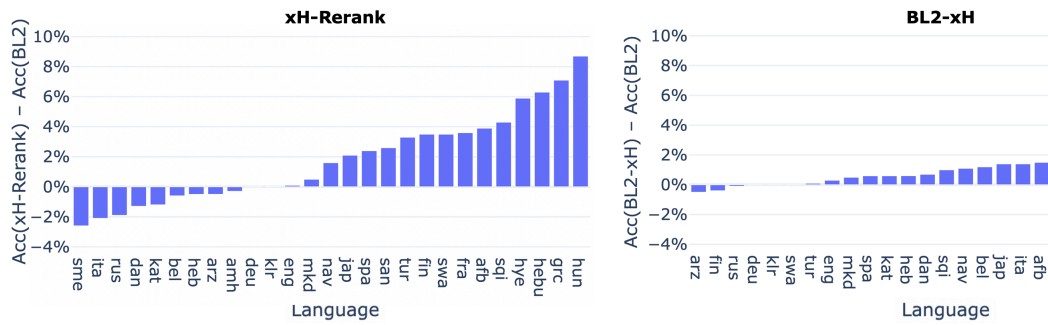

Figure 3: **Accuracy Improvement by Language.** Difference in accuracy between our best models (xH-Rerank and BL2-xH) and our best baseline BL2.

- **xH & MML:** Our bidirectional transformer (§4) trained under the cross-entropy or MML loss of §3.4, and decoded under $P(\boldsymbol{y}, \boldsymbol{o}|\boldsymbol{x})$.

- **xH-Rerank & MML-Rerank:** These variants rerank the 5 candidates returned by beam search of the xH and MML models according to their marginal probability $P(\boldsymbol{y}|\boldsymbol{x})$.

- **BL2-xH & BL2-MML:** These methods select the best L2R or R2L candidate, based on which has higher marginal probability under the xH or MML model.

## 6  Empirical Results

### 6.1  Comparison of Methods

Accuracies averaged over languages are shown in Table 1; results by language are in Appendix D.

**Baselines.** BL2, which selects the higher probability among the L2R and R2L hypotheses, improves by more than 2.3 points in average accuracy over the best unidirectional model. This simple scheme serves as an improved baseline against which to compare our fully bidirectional models.

**xH & MML.** Our bidirectional xH model is clearly more effective than all baselines, having a statistically significant degradation in accuracy on only 3 languages. The MML method is far less effective, beating L2R and R2L but not BL2. MML may suffer from a discrepancy between training and inference, since inference optimizes joint probability while training optimizes likelihood.

**xH- & MML-Rerank.** Reranking according to marginal probability generally improves both bidirectional models. xH-Rerank is the best method overall, beating BL2 by over 1.75 points in average accuracy. MML-Rerank is better than either unidirectional model but still underperforms BL2.

**BL2-xH & BL2-MML.** Selecting the best L2R

or R2L hypothesis based on marginal probability under xH or MML is very effective. Both of these methods improve over BL2, which chooses between the same options based on unidirectional probability. BL2-xH stands out by not having a statistically significant degradation on any language.

**Comparison with Prior SOTA.** Goldman et al. (2023) presents the results of seven other systems submitted to the task; of these, five are from other universities and two are baselines provided by the organizers. The best of these systems is the neural baseline (a unidirectional transformer), which achieves an average accuracy of 81.6 points. Our best system, xH-Rerank, has an accuracy of 84.38 points, achieving an improvement of 2.7 points.

### 6.2  Improvement by Language

Table 1 shows that the best methods are xH-Rerank (by average accuracy) and BL2-xH (improves upon BL2 on the most languages). Figure 3 illustrates this by showing the difference in accuracy between each of these methods and the best baseline BL2.

The plots show that accuracy difference with BL2 has a higher range for xH-Rerank ($-2.6\%$ to $8.7\%$) than for BL2-xH ($-0.5\%$ to $5.8\%$). This is because xH-Rerank has the ability to generate new hypotheses, whereas BL2-xH simply discriminates between the same two hypotheses as BL2.

## 7  Analysis of Results

### 7.1  Length of Output Forms

Figure 2 shows the accuracies by output form length for BL2 and our best method xH-Rerank. xH-Rerank outperforms the baseline at every length (except 10), but especially excels for longer outputs ($\geq 16$ characters). This may be due to the bidirectional model's decreased risk of "snowballing": it can delay the prediction of an uncertain token by

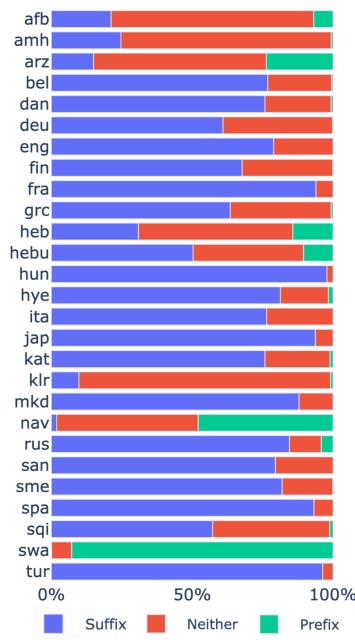
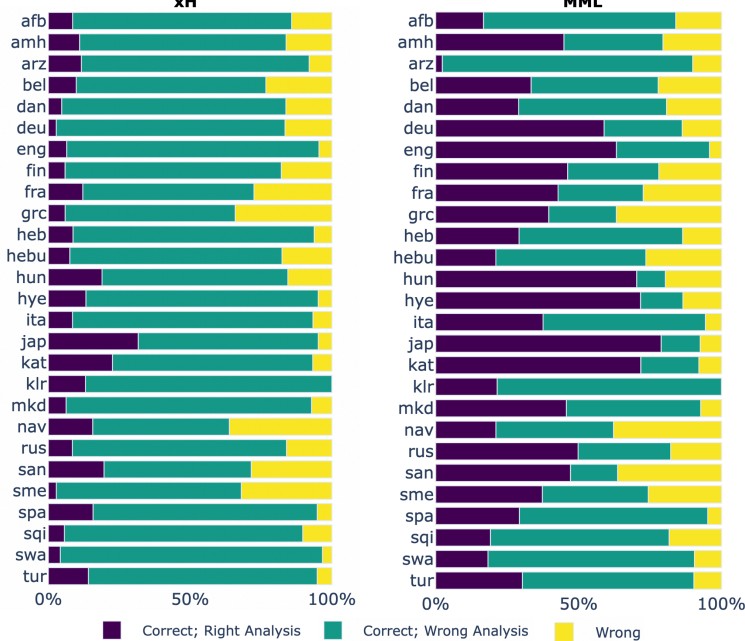

Figure 4: **Morphology of words in test set.** Percentage of forms that are suffix-only, prefix-only, or neither in the test set for each language.

Figure 5: **Analysis for prefix- and suffix-only words.** Percentage of forms for each training method that (1) are correct and whose ordering agrees with the form's morphology; (2) are correct but whose ordering does not agree with the form's morphology; and (3) are incorrect.

generating on the opposite side first, a property not shared with unidirectional models.

## 7.2 How does generation order compare with the morphology of a word?

In this section we consider only forms that can be classified morphologically as prefix-only (e.g. will |**walk**) or suffix-only (e.g. **walk**|ed), because these words have an obvious split point. Ideally, the bidirectional model will exhibit the desired split point by decoding the left and right sides of the form from their respective directions.

We first classify all inflected forms in the test set as suffix-only, prefix-only, or neither. We do this by aligning each lemma-form pair using Levenshtein distance and considering the longest common substring that has length of at least 3 to be the stem.[6] If the inflected form only has an affix attached to the stem, then it is classified as prefix-only or suffix-only; otherwise, it is considered neither.[7]

———————————
[6]For Japanese, we allow the stem to be of length 1 due to the prevalence of Kanji characters in the dataset.

[7]This heuristic approach is likely to work well on examples without infixes or phonetic alternations that obscure the stem. A potential drawback is that it is based on individual lemma-form pairs; a probabilistic method that collects evidence from other examples may be beneficial. However, we feel that the interpretability of this approach as well as qualitative analysis supporting its efficacy makes it sufficient for our study.

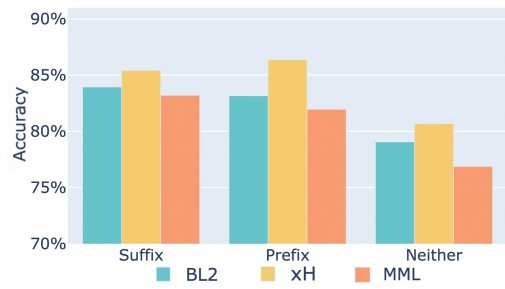

Figure 6: **Accuracy of models by word type.** Accuracy of words that are suffix- or prefix-only, or neither.

Figure 4 shows the percentage of words that are prefix-only, suffix-only, or neither for each language. Most languages favor suffix-only inflections, although Swahili strongly prefers prefixes and several other languages have a high proportion of words without a clear affix.

Finally, Figure 5 shows the percentage of words with a clear affix on which each bidirectional model has the correct analysis. A correct analysis occurs when the model joins the left and right sequences at the correct split point *and* returns the correct word.

It is immediately obvious that the MML models tend to exhibit the correct analysis, while the xH models generally have the wrong analysis. This make sense because MML learns the latent order-

|          | **2022** | | **2023** |
|          | Overall | Unseen | Overall/Unseen |
|----------|---------|--------|----------------|
| **L2R**  | 73.20   | 74.99  | 80.26          |
| **R2L**  | 74.48   | 75.70  | 79.65          |
| **BL2**  | 75.96   | 77.23  | 82.59          |
| **xH**   | 72.76   | 74.85  | 84.25          |
| **xH-Rerank** | 72.91 | 74.72 | 84.38       |
| **BL2-xH** | **76.03** | **78.02** | 84.00     |
| Yang et al. (2022) | 71.26 | 74.96 | –       |

Table 2: **Comparison of 2022 and 2023 results.**
Macro-averaged accuracies over all languages in the
SIGMORPHON 2022 and 2023 shared tasks. Accuracies on test lemmas that are unseen in the training data
are also reported (for 2023, all test lemmas are unseen
in the training data). The average accuracies of the best
system (Yang et al., 2022) submitted to the 2022 shared
task are also reported.

|                    | **Train** | | **Test** | |
|                    | 2022   | 2023  | 2022   | 2023 |
|--------------------|--------|-------|--------|------|
| **Unique Lemma**   | 3636.4 | 753.4 | 1492.0 | 94.1 |
| **Unseen Lemma**   | –      | –     | 619.0  | 94.1 |
| **Forms per Lemma**| 2.5    | 19.3  | 1.4    | 15.4 |
| **Lemmas per Tagset**| 100.9 | 209.9 | 15.4 | 22.1 |

Table 3: **Dataset Statistics.** Number of unique lemmas,
unseen lemmas, average number of forms per lemma,
and average number of lemmas per tagset averaged over
all languages for the 2022 and 2023 datasets. 2022
numbers are scaled to the 2023 size (10k train, ∼1k test
examples) to allow for direct comparison.

ing variable, unlike cross-entropy. Despite MML's
success at learning this morphology, it tends to
have lower accuracy than xH; we explore this by
breaking down accuracy by word type in Figure 6.

Learning the ordering seems to be harmful when
there is no obvious affix: compared with BL2,
MML barely drops in accuracy on prefix- and
suffix-only forms but degrades greatly when there
is no clear split. The xH model, which does not
learn ordering, improves in all categories.

We conclude that MML models better reflect
the stem-affix split than cross-entropy models but
have lower accuracy. Improving the performance
of MML models while maintaining their linguistic
awareness is a promising direction for future work.

### 7.3 Ablation Study: Does bidirectional decoding help?

In this section, we analyze to what extent the bidirectional models' improvement is due to their ability to produce tokens from both sides and meet at
any position. To this end, we force our trained xH
and MML models to decode in a fully L2R or R2L
manner by setting the log probabilities of tokens
in the opposite direction to $-\infty$ at inference time.
The results are shown in Table 4.

The bidirectional models perform poorly when
not permitted to decode from both sides. This is particularly detrimental for the MML model, which is
expected as the marginalized training loss enables
the model to assign low probabilities to some orderings. Clearly, our MML model does not favor
unidirectional orderings.

The xH model, on the other hand, does not suffer
as much from unidirectional decoding. Since it was
trained to treat all orderings equally, we would expect it to do reasonably well on any given ordering.
Nonetheless, it still drops by about 7 points for L2R
decoding and about 13 points for R2L decoding.
This shows that the full bidirectional generation
procedure is crucial to the success of this model.

### 7.4 Results on 2022 Shared Task

We also train our bidirectional cross-entropy model
on the 2022 SIGMORPHON inflection task (Kodner et al., 2022), which, unlike the 2023 data, *does*
have lemmas that occur in both the train and test
sets. The results are shown in Table 2. All of our
methods (including the baselines) outperform the
best submitted system (Yang et al., 2022) on the
2022 data; our best method BL2-xH improves by
over 4.7 points in average accuracy.

However, only BL2-xH outperforms the baseline
BL2 (barely), which is in stark contrast to the 2023
task, where all cross-entropy-based methods beat
the baseline considerably. To make the comparison
between the years more fair, we evaluate the 2022
models only on lemmas in the test set that did not
occur in training. Again, only BL2-xH outperforms
the baseline, this time by a wider margin; xH and
xH-Rerank still underperform.

We posit that this discrepancy is likely due to the
considerably different properties of the 2022 and
2023 datasets, which are shown in Table 3. The
2023 languages have far fewer unique lemmas and
have many more forms per lemma. Hence, it seems
that our bidirectional model improves much more
compared with the baseline when there are fewer
but more "complete" paradigms.

This investigation shows that the performance of
inflection models depends substantially on the data
sampling, which is not always controlled for. Kodner et al. (2023) makes progress on this matter, but

| | Bidi | Forced L2R | Forced R2L | Bidi-2 |
|---|---|---|---|---|
| **Uni** | – | 80.26 | 79.65 | 82.59 |
| **xH** | 84.25 | 71.05 | 77.31 | 78.42 |
| **MML** | 81.43 | 4.68 | 0.07 | 2.33 |

Table 4: **Ablation study on 2023 dataset.** Macro-averaged accuracies for bidirectional models decoded using the method of §3.5 (Bidi), or when forced to decode in an L2R or R2L manner. Bidi-2 indicates the outcome when selecting between the forced unidirectional decodings based on which has a higher probability. The unidirectional models (Uni) indicate the accuracies of standard unidirectional transformers and BL2.

does not explicitly examine paradigm "completeness", which should be a focus in future studies.

## 8 Conclusion

We have proposed a novel framework for bidirectional decoding that allows a model to choose the generation order for each sequence, a major difference from previous work. Further, our method enables an efficient dynamic programming algorithm for training, which arises due to an independence assumption that can be built into our transformer-based architecture. We also present a simple beam-search algorithm for decoding, the outputs of which can optionally be reranked using the likelihood calculation. Our model beats SOTA on both the 2022 and 2023 shared tasks without resorting to data augmentation. Further investigations show that our model is especially effective on longer output words and can implicitly learn the morpheme boundaries of output sequences.

There are several avenues for future research. One open question is the extent to which data augmentation can improve accuracy. We also leave open the opportunity to explore our bidirectional framework on other sequence tasks, such as machine translation, grapheme-to-phoneme conversion, and named-entity transliteration. Various other architectures could also be investigated, such as the bidirectional attention mechanism of Zhou et al. (2019b) or non-transformer based approaches. Finally, given the effectiveness of MML reranking, it could be worthwhile to explore efficient approaches to decode using marginal probability.

## 9 Limitations

We acknowledge several limitations to our work. For one, we only demonstrate experiments on the inflection task, which is fairly straightforward in some ways: there is typically only one output for a given input (unlike translation, for example), and a large part of the output is copied from the input. It would be informative to test the efficacy of our bidirectional framework on more diverse generation tasks, such as translation or question-answering.

From a practical standpoint, the most serious limitation is that, in order to use dynamic programming, the model architecture cannot be trained with a causal mask: all hidden states must be recomputed at each timestep. Further, our xH and MML schemes are quadratic in sequence length. These two properties cause the training time of our bidirectional method to be $O(|\boldsymbol{y}|^4)$ in runtime rather than $O(|\boldsymbol{y}|^2)$ (like the standard transformer).[8] Alleviating these constraints would enable a wider variety of experiments on tasks with longer sequences.

## Acknowledgements

This work utilizes resources supported by the National Science Foundation's Major Research Instrumentation program, grant #1725729, as well as the University of Illinois at Urbana-Champaign. In particular, we made significant use of the HAL computer system (Kindratenko et al., 2020). We would also like to acknowledge Weights & Biases (Biewald, 2020), which we utilized to manage our experiments.

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

## A    Datasets, Hyperparameter Tuning, & Validation Accuracies

The languages in the SIGMORPHON 2022 and 2023 datasets are listed in Tables 7 and 8. We experiment with small, medium, and large model sizes for each language, whose configurations and approximate number of parameters can be found in Table 5:

| | S | M | L |
|---|---|---|---|
| Embed dim | 64 | 128 | 256 |
| FFN dim | 256 | 512 | 1024 |
| Num. layers | 2 | 3 | 4 |
| Num. heads | 2 | 4 | 8 |
| Learning rate | 0.005 | 0.001 | 0.001 |
| Num. params | $\sim 240k$ | $\sim 1.4M$ | $\sim 7.3M$ |

Table 5: **Hyperparameters.** Hyperparameters for small, medium, and large models.

For each language, we train L2R and R2L models (with random initialization) for each hyperparameter size (a total of 6 models per language), and select a size based on the average of the L2R and R2L validation accuracies. The model sizes chosen for each language, along with each language's validation accuracies, are reported in Tables 13 and 14.

Note that the number of parameters vary slightly among languages due to different vocabulary sizes (i.e. number of unique characters in the training set), and the bidirectional models also have a small number of extra parameters due to the additional classification tokens described in §4.

## B    Handling Unknown Characters

If an unknown character is encountered in a lemma at test time, then a special UNK character is used; however, this character is not explicitly trained. If an UNK character is predicted by the model, then we replace it with the first (leftmost) unknown character in the lemma; if no such character exists then it is ignored.

We adopt a special scheme for Japanese, which has a very high number of unknown characters. All characters that occur fewer than 100 times in the training set are considered "unknown". If a lemma has $n$ unknown tokens, then these are replaced with $UNK_1$, ..., $UNK_n$; the corresponding tokens in the inflected form are replaced as well. In this way, the model can learn to copy rare or unknown characters to their appropriate locations in the output. At test time, each predicted unknown token is replaced with its corresponding character in the lemma.

## C    Tempering the Order Distribution at Train Time

Initial empirical results showed that training with MML loss caused the model to quickly reach a "degenerate" state, where every sequence was decoded in the same direction. To encourage the model to explore different orderings at an early stage, we *temper* the order probabilities over a warmup period. The temperature is degraded from initial temperature $\tau_0$ to $1$ over a period of $W$ steps as follows:

$$\tau_n = \frac{\tau_0 - 1}{W^a}(W - n)^a + 1$$

The parameter $a$ controls how fast the shift occurs, and $n$ corresponds to the training step. This temperature is applied to the softmax of order probabilities for the first $W$ steps of training.

In our experiments, we set $W = 4,000$, $\tau_0 = 50$ and $a = 2$.

## D    All Results

The accuracies for all languages in our study are shown in Table 9 (2023 data) and Table 10 (2022 data). These tables also display L2R-Rerank (which reranks the 5 candidates from the L2R model's beam search under the cross-entropy or MML model), R2L-Rerank, and (L2R+R2L)-Rerank (which reranks the 10 candidates returned from the L2R and R2L's beam search under the cross-entropy or MML model).

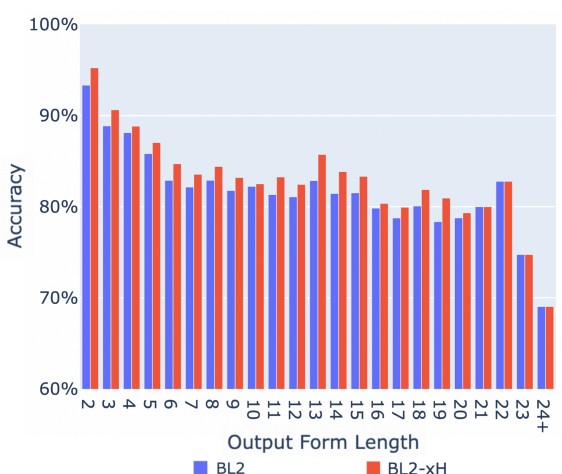

Figure 7: **Accuracies of BL2-xH and BL2 by Word Length.** Average accuracies of BL2 and BL2-xH models over all languages, grouped by length (number of characters) of the output form.

## E Oracle Scores

Table 11 shows the oracle score for each method; this gives an upper bound for choosing among a set of hypotheses. We see that both xH-Rerank and BL2-Rerank approach their respective bounds: the average accuracy for xH-Rerank is within 1 point of its oracle score, and the average accuracy for BL2-xH is within 2 points of its oracle score.

## F Cross-entropy with Random Path (xH-Rand)

The cross-entropy loss presented in §3.4 requires enumerating all $O(|\boldsymbol{y}|^2)$ prefix-suffix pairs. Here, we propose an $O(|\boldsymbol{y}|)$ variant in which the join loss is averaged over a *random* set of prefix-suffix pairs for each word. Specifically, the set $\mathcal{S}$ is defined such that there is only one $(i, j)$ pair for each $1 \leq k \leq |\boldsymbol{y}|$ where $i + j = k$. Otherwise, this loss $\mathcal{L}_{xH\text{-}Rand}(\theta)$ is the same as the cross-entropy loss of §3.4. Since this loss has an $O(|\boldsymbol{y}|)$ runtime, it has the same complexity as a standard unidirectional loss (assuming all local probabilities take constant time to compute).

Table 12 compares the accuracies of this model with the other bidirectional variants discussed in §6. Reranking xH-Rand is slightly better than not reranking, and this performs well: its average accuracy is almost 1 percentage point higher than BL2 and it improves on 15/27 languages. xH-Rand is better than MML but not as good as xH. Nonetheless, its faster runtime and competitive performance makes this a useful method.

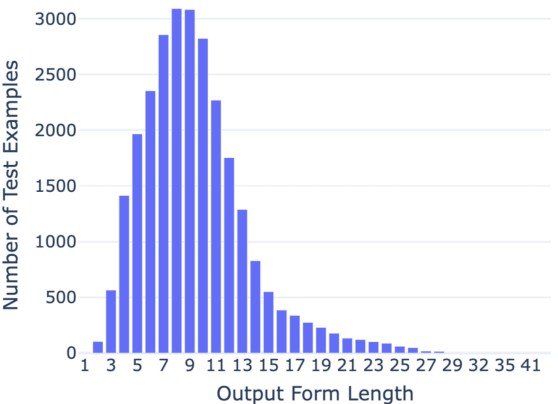

Figure 8: **Number of Test Examples by Length.** Number of test examples across all languages by number of characters in (correct) output form.

## G Additional Results

### G.1 Accuracy by Length

Figure 2 in §7.1 compares the accuracy of our bidirectional method xH-Rerank with that of the baseline BL2 by the length of the output form. Figure 7 shows a similar comparison for BL2-xH (our other best method) with BL2; consistent with the analysis of §6.2, there is less of a difference between these methods, but BL2-xH does equal or outperform BL2 at all lengths.

Figure 8 shows the distribution of output form length across all languages.

### G.2 Accuracy by Part-of-Speech

Figures 10 and 9 compare the accuracies of xH-Rerank and BL2-xH (our best bidirectional methods) with the accuracy of BL2 by part-of-speech. We see that xH-Rerank maintains or improves accuracy over BL2 in all categories except V.MSDR (masdars), and BL2-xH maintains or improves accuracy in all categories except V.MSDR and V.PTCP (participles). These categories make up a small fraction of the data; this can be seen in Figure 11, which shows the distribution of part-of-speech categories across all languages.

### G.3 What orderings does each method prefer?

In this section, we investigate the ordering preferences for each method: does a model prefer to decode words entirely in the L2R or R2L direction, or partially in each direction? These results can be seen for each language in Figure 12.

Both the xH and MML methods have a strong tendency to decode words partially in each di-

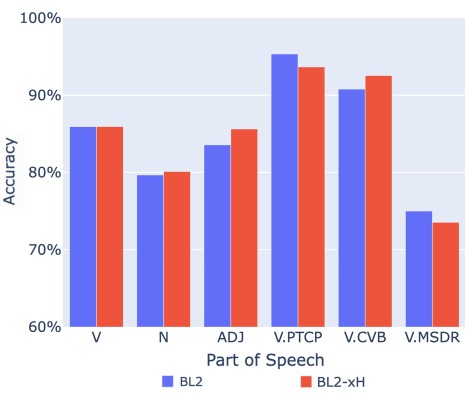

Figure 9: **Accuracies of BL2-xH and BL2 by Part of Speech.** Accuracies of BL2 and BL2-xH models averaged over all languages, grouped by part of speech.

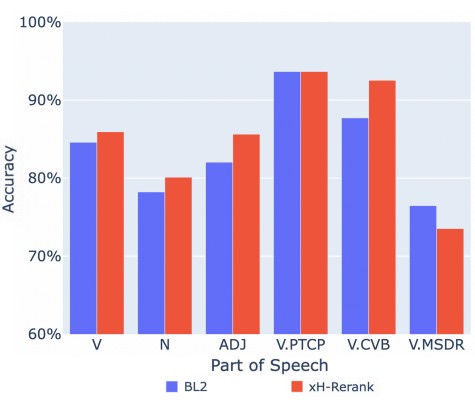

Figure 10: **Accuracies of xH-Rerank and BL2 by Part of Speech.** Accuracies of BL2 and xH-Rerank models averaged over all languages, grouped by part of speech.

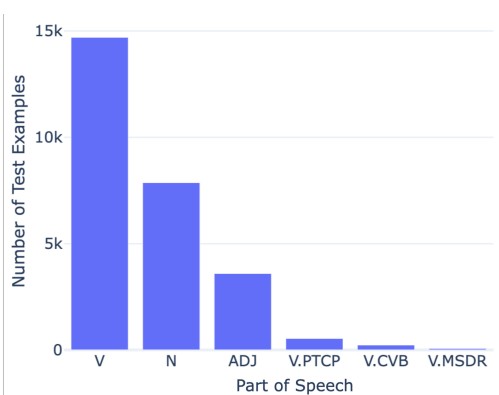

Figure 11: **Number of Test Examples by Part-of-speech.** Number of test examples across all languages by part-of-speech.

|  | Time per 50 examples (s) |
|---|---|
| **L2R** | 0.276 |
| **R2L** | 0.277 |
| **xH** | 0.724 |
| **xH-Rand** | 0.738 |
| **MML** | 0.772 |

Table 6: **Inference times.** Average time to perform inference on 50 test examples averaged over all languages on the 2023 dataset on 4 NVIDIA V100 GPU's.

rection; however, MML models clearly have a higher proportion of words decoded from both directions than their xH counterparts. Out of the words decoded entirely in one direction, the xH model shows a slight preference for R2L generations, though most languages have words decoded from both directions. On the other hand, for the MML model, no language shows a preference for R2L generations over L2R generations; in fact, R2L generations are extremely rare for the MML models.

### G.4 Empirical Inference Times

Given that our bidirectional model must recompute previous hidden states at each timestep during inference (see §4), we wish to compare the empirical slowdown in decoding for our bidirectional models compared with unidirectional models. The average number of seconds taken to decode 50 examples is shown in Table 6.

Recomputing hidden states at each step slows down inference by a factor of about 3. However, in practice, we barely notice the difference on this task, as the test sets have only 1,000 examples each. Given the strong outperformance of the bidirectional methods over the unidirectional baselines (and even over the naive bidirectional baseline BL2), one must therefore make a tradeoff between time and performance.

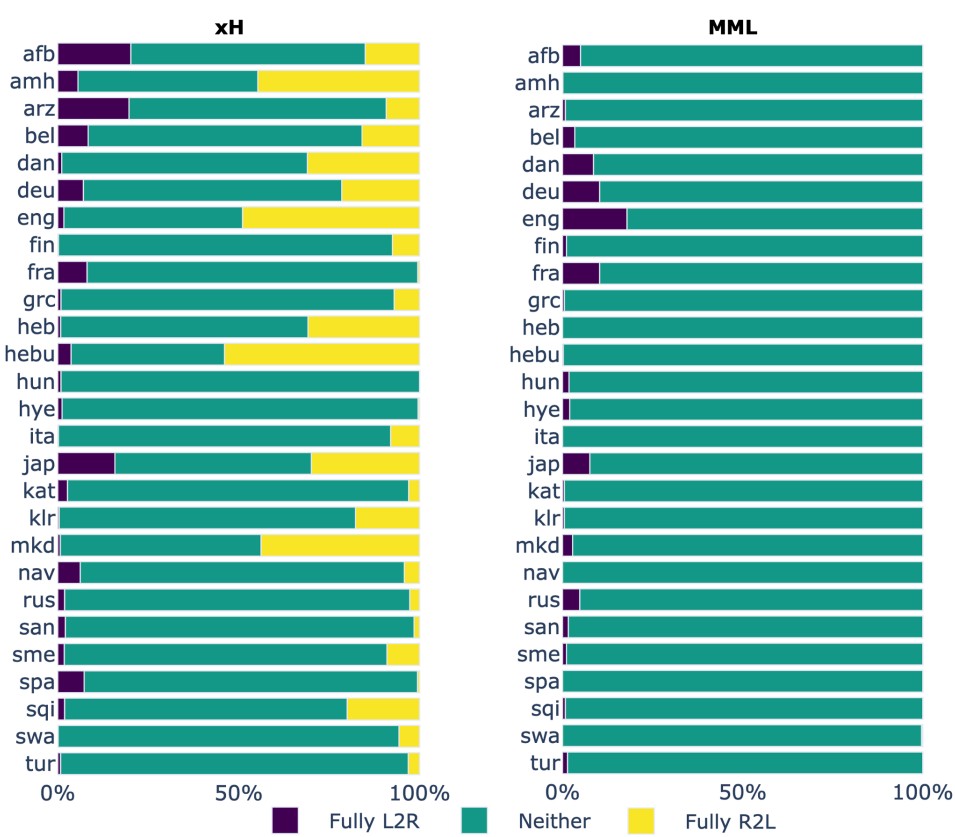

Figure 12: **Ordering choices.** Percentage of examples for each language and training method that are decoded fully L2R, fully R2L, or partially from each direction.

Table 7: **2023 Dataset Information.** Information on each language in the 2023 dataset, including language family and genus, baseline accuracies on test set, and model size chosen (based on validation accuracies).

| Language | Family | Genus | Language Code | L2R Test Accuracies | | | R2L Test Accuracies | | | Model Size Chosen |
|---|---|---|---|---|---|---|---|---|---|---|
| | | | | Small | Medium | Large | Small | Medium | Large | |
| **Gulf Arabic** | Afro-Asiatic | Semitic | afb | 75.20 | 74.70 | 75.10 | 78.10 | 76.10 | 75.50 | small |
| **Amharic** | Afro-Asiatic | Semitic | amh | 86.20 | 84.40 | 83.50 | 82.80 | 89.30 | 83.20 | medium |
| **Egyptian Arabic** | Afro-Asiatic | Semitic | arz | 87.20 | 89.60 | 87.80 | 88.10 | 87.30 | 86.70 | small |
| **Belarusian** | Indo-European | Slavic | bel | 71.70 | 73.00 | 70.30 | 68.80 | 74.80 | 70.10 | large |
| **Danish** | Indo-European | Germanic | dan | 87.70 | 88.40 | 87.10 | 86.20 | 87.80 | 86.50 | medium |
| **German** | Indo-European | Germanic | deu | 80.80 | 75.70 | 73.10 | 76.50 | 77.10 | 78.00 | large |
| **English** | Indo-European | Germanic | eng | 94.60 | 94.50 | 94.80 | 93.80 | 94.00 | 93.20 | medium |
| **Finnish** | Uralic | Finnic | fin | 81.60 | 78.30 | 76.90 | 72.60 | 74.00 | 72.80 | medium |
| **French** | Indo-European | Romance | fra | 63.70 | 65.10 | 57.20 | 67.70 | 59.50 | 50.10 | small |
| **Ancient Greek** | Indo-European | Hellenic | grc | 49.40 | 53.20 | 50.80 | 47.50 | 39.10 | 34.80 | medium |
| **Hebrew** | Afro-Asiatic | Semitic | heb | 87.41 | 93.25 | 91.14 | 88.22 | 90.33 | 87.41 | large |
| **Hebrew (Unvocalized)** | Afro-Asiatic | Semitic | hebu / heb_unvoc | 79.40 | 78.50 | 76.50 | 71.40 | 74.10 | 73.40 | medium |
| **Hungarian** | Uralic | Ugric | hun | 77.70 | 77.40 | 73.30 | 66.10 | 71.10 | 70.40 | small |
| **Armenian** | Indo-European | Armenian | hye | 82.00 | 86.00 | 84.00 | 86.40 | 76.90 | 65.70 | small |
| **Italian** | Indo-European | Romance | ita | 89.30 | 82.90 | 91.40 | 93.90 | 78.60 | 79.80 | small |
| **Japanese** | Japonic | — | jap | 93.10 | 93.80 | 91.90 | 89.60 | 91.00 | 92.60 | medium |
| **Georgian** | Kartvelian | Karto-Zan | kat | 79.70 | 76.30 | 76.60 | 79.50 | 82.00 | 69.00 | small |
| **Khaling** | Sino-Tibetan | Kiranti | klr | 98.30 | 99.40 | 95.80 | 95.60 | 98.30 | 98.30 | medium |
| **Macedonian** | Indo-European | Slavic | mkd | 90.90 | 89.70 | 89.70 | 89.40 | 92.00 | 88.80 | medium |
| **Navajo** | Na-Dené | Southern Athabaskan | nav | 53.70 | 53.40 | 45.60 | 48.90 | 53.30 | 52.60 | small |
| **Russian** | Indo-European | Slavic | rus | 82.10 | 88.20 | 84.00 | 84.90 | 86.00 | 80.10 | small |
| **Sanskrit** | Indo-European | Indic | san | 61.50 | 52.40 | 41.50 | 60.70 | 44.10 | 44.00 | small |
| **Sami** | Uralic | Finnic | sme | 64.10 | 63.40 | 56.50 | 61.10 | 65.30 | 56.40 | medium |
| **Spanish** | Indo-European | Romance | spa | 90.40 | 90.30 | 88.90 | 93.50 | 91.20 | 87.50 | medium |
| **Albanian** | Indo-European | Albanian | sqi | 82.50 | 85.00 | 80.00 | 88.30 | 84.40 | 75.00 | medium |
| **Swahili** | Niger-Congo | Bantu | swa | 89.00 | 92.70 | 86.90 | 92.90 | 92.90 | 92.40 | medium |
| **Turkish** | Turkic | Oghuz | tur | 88.80 | 88.80 | 89.10 | 87.40 | 83.20 | 80.00 | small |

| Language | Linguistic Information | | Language Code | L2R Test Accuracies | | | R2L Test Accuracies | | | Model Size Chosen |
|---|---|---|---|---|---|---|---|---|---|---|
| | Family | Genus | | Small | Medium | Large | Small | Medium | Large | |
| **Old English** | Indo-European | Germanic | ang | 56.78 | 59.93 | 62.06 | 60.49 | 63.03 | 64.55 | large |
| **Arabic** | Afro-Asiatic | Semitic | ara | 77.74 | 76.94 | 77.49 | 77.84 | 77.59 | 77.24 | small |
| **Assamese** | Indo-European | Indic | asm | 81.46 | 80.75 | 81.11 | 74.42 | 84.87 | 83.57 | medium |
| **Evenki** | Tungusic | Northern Tungusic | evn | 56.11 | 56.05 | 54.68 | 57.37 | 54.79 | 52.09 | small |
| **Gothic** | Indo-European | Germanic | got | 72.52 | 73.47 | 73.62 | 70.11 | 68.51 | 74.77 | large |
| **Hebrew** | Afro-Asiatic | Semitic | heb | 47.90 | 49.50 | 49.35 | 52.70 | 50.55 | 53.10 | medium |
| **Hungarian** | Uralic | Ugric | hun | 68.20 | 78.40 | 75.80 | 72.65 | 76.40 | 75.80 | medium |
| **Armenian** | Indo-European | Armenian | hye | 89.90 | 91.90 | 91.15 | 90.15 | 92.90 | 92.70 | medium |
| **Georgian** | Kartvelian | Karto-Zan | kat | 84.15 | 86.10 | 84.50 | 83.85 | 88.15 | 88.95 | large |
| **Kazakh** | Turkic | Kipchak | kaz | 64.14 | 64.34 | 62.99 | 62.54 | 69.11 | 68.15 | medium |
| **Khalkha Mongolian** | Mongolic | — | khk | 46.26 | 48.59 | 48.69 | 46.97 | 48.59 | 48.43 | medium |
| **Korean** | Koreanic | — | kor | 57.13 | 57.18 | 57.54 | 57.23 | 58.76 | 57.64 | large |
| **Karelian** | Uralic | Finnic | krl | 65.18 | 71.49 | 67.94 | 70.79 | 70.89 | 72.39 | medium |
| **Ludic** | Uralic | Finnic | lud | 63.16 | 74.14 | 68.27 | 72.32 | 78.34 | 56.98 | medium |
| **Old Norse** | Indo-European | Germanic | non | 82.92 | 84.43 | 84.73 | 82.92 | 82.32 | 85.89 | large |
| **Polish** | Indo-European | Slavic | pol | 90.10 | 90.40 | 88.95 | 89.30 | 89.15 | 89.90 | medium |
| **Pomak** | Indo-European | Slavic | poma | 66.98 | 65.68 | 67.83 | 66.88 | 64.23 | 67.88 | large |
| **Slovak** | Indo-European | Slavic | slk | 91.90 | 92.55 | 93.25 | 93.25 | 94.10 | 93.45 | large |
| **Turkish** | Turkic | Oghuz | tur | 94.15 | 93.85 | 93.70 | 93.65 | 95.40 | 92.90 | medium |
| **Veps** | Uralic | Finnic | vep | 60.91 | 61.26 | 63.37 | 60.56 | 60.41 | 62.57 | large |

Table 8: **2022 Dataset Information.** Information on each language in the 2022 dataset, including language family and genus, baseline accuracies on test set, and model size chosen (based on validation accuracies). Only large datasets (7,000 train examples) are used.

Table 9 — **All Accuracies (2023 data).**

| Model | Size | Baselines | | | Standalone | | xH-Rerank | | MML-Rerank | | BL Discriminator | | L2R-Rerank | | R2L-Rerank | | (L2R+R2L)-Rerank | |
|---|---|---|---|---|---|---|---|---|---|---|---|---|---|---|---|---|---|---|
| | | L2R | R2L | BL2 | xH | MML | xH | MML | xH | MML | xH | MML | xH | MML | xH | MML | xH | MML |
| afb | small | 75.20 | 78.10 | 80.70 | 84.10* | 78.70* | 84.60* | 84.90* | 81.50 | 79.20 | 82.20* | 80.30 | 80.20 | 78.00* | 80.00 | 76.90* | 82.90* | 79.20 |
| amh | medium | 84.40 | 89.30 | 88.90 | 88.90 | 83.40* | 88.60 | 87.90 | 85.90* | 83.40* | 90.60* | 87.30* | 85.80* | 82.90* | 89.30 | 84.70* | 89.10 | 83.60* |
| arz | small | 87.20 | 88.10 | 89.20 | 89.10 | 87.50 | 88.70 | 88.90 | 87.30* | 87.40* | 88.70 | 89.10 | 89.20 | 88.50 | 88.30 | 89.20 | 89.10 | 88.90 |
| bel | large | 70.30 | 70.10 | 73.50 | 72.90 | 72.80 | 72.90 | 73.20 | 74.00 | 72.90 | 74.70* | 74.40 | 71.80 | 73.60 | 71.60 | 73.80 | 72.10 | 74.10 |
| dan | medium | 88.40 | 87.80 | 88.80 | 86.50* | 83.60* | 87.50 | 86.80* | 85.00* | 83.60* | 89.50 | 89.60 | 89.30 | 86.80* | 86.80* | 85.70* | 88.30 | 85.40* |
| deu | large | 73.10 | 78.00 | 79.70 | 80.20 | 81.10 | 79.70 | 80.80 | 80.80 | 81.00 | 79.70 | 81.00* | 74.80* | 75.70* | 80.30 | 81.50* | 80.20 | 81.90* |
| eng | medium | 94.50 | 94.00 | 95.60 | 95.70 | 95.80 | 95.70 | 96.00 | 96.00 | 95.80 | 95.90 | 95.80 | 95.50 | 95.60 | 94.90 | 95.50 | 95.50 | 95.80 |
| fin | medium | 78.30 | 74.00 | 79.20 | 83.60* | 77.70 | 82.70* | 83.90* | 82.70* | 77.90 | 81.20* | 81.20* | 81.20* | 81.00 | 78.10 | 78.10 | 81.20* | 79.80 |
| fra | small | 63.70 | 67.70 | 69.30 | 71.70 | 71.60 | 72.90* | 73.50* | 74.90* | 71.50 | 74.70* | 74.40* | 72.00* | 70.70 | 72.70* | 74.00* | 75.00* | 75.00* |
| grc | medium | 53.20 | 39.10 | 48.90 | 56.00* | 53.50* | 56.00* | 55.70* | 53.60* | 53.30* | 53.70* | 55.10* | 54.60* | 55.80* | 41.50* | 43.10* | 59.30* | 59.30* |
| heb | large | 91.14 | 87.41 | 92.95 | 92.45 | 84.09* | 92.45 | 92.25 | 86.10* | 84.09* | 93.55 | 91.04* | 91.74 | 90.53* | 88.62* | 84.99* | 91.64 | 86.20* |
| hbo | medium | 78.50 | 74.10 | 77.30 | 83.70* | 75.00 | 83.60* | 83.60* | 81.50* | 75.00 | 79.30* | 78.20 | 79.30* | 77.50 | 79.40* | 74.20* | 82.40* | 76.20 |
| hun | small | 77.70 | 66.10 | 76.30 | 84.30* | 79.30* | 85.00* | 85.10* | 82.30* | 80.10* | 79.80* | 81.20* | 81.70* | 79.10* | 75.20 | 76.10 | 81.10* | 81.10* |
| hye | small | 82.00 | 86.40 | 88.40 | 94.20* | 86.50 | 94.30* | 94.20* | 88.40 | 86.20 | 91.40* | 91.10* | 85.10* | 81.50* | 88.80 | 87.60 | 92.70* | 88.00 |
| ita | small | 89.30 | 93.90 | 95.80 | 94.40 | 92.70* | 94.90* | 92.10* | 95.30 | 92.70* | 97.20* | 96.40 | 90.10* | 90.60* | 93.90* | 95.40 | 94.30* | 94.70 |
| jap | medium | 93.80 | 91.00 | 92.80 | 92.30 | 92.30 | 92.90 | 94.20 | 93.50 | 92.10 | 94.20* | 93.40 | 94.80* | 93.00 | 92.20 | 94.70 | 92.40 | 94.70 |
| kat | small | 79.70 | 79.50 | 84.10 | 81.30* | 81.40* | 82.90 | 82.60 | 83.50 | 81.10* | 84.70* | 84.60 | 82.70* | 84.30 | 81.40* | 83.70 | 83.70 | 92.40 |
| klr | medium | 99.40 | 98.30 | 99.40 | 99.40 | 99.40 | 99.40 | 99.40 | 99.40 | 99.40 | 99.40 | 99.40 | 99.40 | 99.40 | 99.00 | 99.00 | 99.40 | 99.40 |
| mkd | medium | 89.70 | 92.00 | 91.90 | 92.10 | 91.40 | 92.40 | 92.40 | 93.20 | 91.50 | 92.40 | 91.90 | 91.90 | 92.00 | 93.20 | 92.70 | 91.90 | 91.90 |
| nav | small | 53.70 | 48.90 | 54.00 | 55.10 | 57.10* | 55.60 | 55.00 | 57.00* | 57.00* | 55.10 | 55.60* | 54.90 | 56.40* | 54.30 | 55.60 | 56.00 | 57.10* |
| rus | small | 82.10 | 84.90 | 87.40 | 84.20* | 82.10* | 85.50* | 85.40* | 84.70* | 83.30* | 87.30 | 87.30 | 83.30* | 81.20* | 86.30 | 86.00 | 84.30* | 84.30* |
| san | medium | 61.50 | 60.70 | 63.30 | 67.70* | 59.60* | 65.90 | 66.40* | 66.10* | 60.60 | 69.10* | 68.80* | 67.10* | 65.50 | 61.90 | 60.50* | 66.60* | 65.20 |
| sme | medium | 63.40 | 65.30 | 69.90 | 67.40 | 75.60* | 67.30 | 69.00 | 70.00 | 75.20* | 71.80* | 70.90 | 66.60* | 70.00 | 67.90 | 71.10 | 69.90 | 75.00* |
| spa | medium | 90.30 | 91.20 | 90.90 | 93.20* | 93.40* | 93.30* | 93.30* | 93.30* | 93.50* | 91.50 | 91.00 | 91.50 | 91.80 | 91.80 | 91.70 | 92.30* | 92.00* |
| sqi | medium | 85.00 | 84.40 | 87.60 | 91.00* | 82.50* | 91.90* | 89.90 | 82.80* | 82.40* | 88.60* | 85.90* | 89.30 | 85.80* | 87.90 | 85.30* | 90.30* | 86.60 |
| swa | medium | 92.70 | 92.90 | 93.10 | 96.60* | 90.50* | 96.60* | 95.60* | 91.40* | 90.50* | 93.10 | 93.00 | 93.10 | 92.80 | 92.90 | 92.90 | 93.50 | 93.00 |
| tur | small | 88.80 | 87.40 | 90.90 | 94.00* | 89.90 | 94.20* | 93.40* | 93.00* | 89.90 | 91.00 | 90.30 | 92.40* | 91.60 | 88.10* | 86.10* | 93.30* | 91.30 |
| **Average Number** | | 80.26 | 79.65 | 82.59 | 84.25 | 81.43 | 84.38 | 84.26 | 82.90 | 81.50 | 84.00 | 83.54 | 82.64 | 81.85 | 81.81 | 81.29 | 84.11 | 83.00 |
| **Number** ($p \leq 0.05$) > | | | | | 19/27 | 9/27 | 18/27 | 18/27 | 17/27 | 9/27 | 24/27 | 18/27 | 16/27 | 16/27 | 11/27 | 11/27 | 19/27 | 14/27 |
| **Number** ($p \leq 0.05$) = | | | | | 5/27 | 7/27 | 7/27 | 7/27 | 8/27 | 6/27 | 3/27 | 6/27 | 4/27 | 2/27 | 10/27 | 1/27 | 7/27 | 9/27 |
| **Number** ($p \leq 0.05$) < | | | | | 3/27 | 11/27 | 2/27 | 2/27 | 2/27 | 12/27 | 0/27 | 3/27 | 7/27 | 9/27 | 6/27 | 15/27 | 1/27 | 4/27 |

Table 9: **All Accuracies (2023 data).** A number is starred (*) if it shows a statistically significant difference with the best baseline BL2; a number is colored in green if it improves over BL2 (regardless of significance) using a paired permutation test (Zmigrod et al., 2022); and a number is **bold** if it is the best for the language.

| Model | Size | Baselines | | | Bidirectional | | Baselines Rerank | | | |
|---|---|---|---|---|---|---|---|---|---|---|
| | | L2R | R2L | BL2 | xH | xH-Rerank | BL2-xH | L2R-Rerank-xH | R2L-Rerank-xH | (L2R+R2L)-Rerank-xH |
| ang | large | 62.06 | 64.55 | 65.26 | 60.89 | 61.05 | **65.77** | 62.77 | 64.20 | 63.53 |
| ara | small | 77.74 | 77.84 | **79.45** | 76.19 | 77.09 | 79.20 | 78.70 | 77.59 | 78.45 |
| asm | medium | 80.75 | 84.87 | 85.73 | 83.42 | 83.22 | **87.44** | 83.87 | 83.67 | 85.18 |
| evn | small | 56.11 | 57.37 | 60.70 | 56.86 | 56.86 | **61.10** | 56.51 | 58.00 | 58.35 |
| got | large | 73.62 | 74.77 | 75.33 | 72.17 | 72.62 | **75.83** | 74.32 | 74.02 | 74.07 |
| heb | large | 49.35 | 53.10 | **53.95** | 49.95 | 49.95 | 52.00 | 49.35 | 50.80 | 50.35 |
| hun | medium | **78.40** | 76.40 | 78.15 | 77.00 | 77.00 | 77.80 | 77.60 | 76.60 | 77.05 |
| hye | medium | 91.90 | 92.90 | 93.60 | 90.30 | 90.85 | **93.90** | 92.10 | 92.60 | 92.05 |
| kat | large | 84.50 | 88.95 | 88.95 | **92.00** | 91.50 | 89.75 | 87.70 | 90.30 | 91.70 |
| kaz | medium | 64.34 | 69.11 | 70.36 | 65.70 | 66.70 | **70.96** | 63.19 | 69.46 | 69.01 |
| khk | medium | 48.59 | 48.59 | 48.94 | 48.94 | 48.99 | **48.99** | 48.79 | 48.94 | **48.99** |
| kor | large | 57.54 | 57.64 | 59.11 | 57.64 | 57.69 | **59.93** | 57.59 | 58.81 | 58.81 |
| krl | medium | 71.49 | 70.89 | **73.40** | 64.38 | 65.93 | 72.34 | 69.59 | 68.99 | 68.49 |
| lud | medium | 74.14 | 78.34 | **82.19** | 64.37 | 63.82 | 80.62 | 70.34 | 77.13 | 71.26 |
| non | large | 84.73 | 85.89 | 87.95 | 84.88 | 85.03 | **88.05** | 84.83 | 85.84 | 85.64 |
| pol | medium | 90.40 | 89.15 | **90.95** | 89.40 | 89.15 | 90.80 | 89.70 | 89.40 | 89.70 |
| poma | large | 67.83 | 67.88 | 69.78 | 67.88 | 67.93 | **70.14** | 69.73 | 68.83 | 69.03 |
| slk | large | 93.25 | 93.45 | 94.20 | 94.05 | 94.00 | **94.90** | 93.55 | 93.10 | 94.15 |
| tur | medium | 93.85 | 95.40 | 95.75 | 95.60 | 95.60 | 95.60 | 95.20 | **95.80** | 95.75 |
| vep | large | 63.37 | 62.57 | 65.43 | 63.67 | 63.32 | **65.48** | 64.33 | 63.87 | 64.53 |
| **Average** | | 73.20 | 74.48 | 75.96 | 72.76 | 72.91 | **76.03** | 73.49 | 74.40 | 74.30 |
| **Number ≥** | | | | | 2/20 | 2/20 | **13/20** | 0/20 | 3/20 | 3/20 |

Table 10: **All Accuracies (2022 data).** A number is colored in green if it improves over BL2, and a number is **bold** if it is the best for the language.

|  | Model | Baselines | | | | Bidirectional | | |
|---|---|---|---|---|---|---|---|---|
|  | Size | L2R | R2L | BL2 | BL10 | xH | xH-Rand | MML |
| afb | small | 89.00 | 89.50 | 85.10 | 93.40 | 86.20 | 83.80 | 84.90 |
| amh | medium | 89.30 | 96.40 | 92.00 | 96.70 | 89.00 | 89.00 | 89.20 |
| arz | small | 94.90 | 94.70 | 90.70 | 96.30 | 89.20 | 88.50 | 89.60 |
| bel | large | 82.60 | 82.40 | 77.40 | 87.30 | 75.10 | 74.70 | 78.50 |
| dan | medium | 95.00 | 92.40 | 92.50 | 97.00 | 88.50 | 89.10 | 87.80 |
| deu | large | 81.60 | 86.50 | 82.30 | 90.30 | 81.20 | 81.30 | 83.80 |
| eng | medium | 98.20 | 97.10 | 96.70 | 98.70 | 96.40 | 96.70 | 96.90 |
| fin | medium | 89.50 | 83.60 | 81.40 | 91.00 | 84.50 | 86.40 | 80.40 |
| fra | small | 86.00 | 89.60 | 79.40 | 94.90 | 74.40 | 73.20 | 80.90 |
| grc | medium | 63.00 | 48.50 | 55.80 | 69.90 | 56.00 | 49.10 | 55.50 |
| heb | large | 95.07 | 90.43 | 94.36 | 96.68 | 92.45 | 89.43 | 86.30 |
| hebu | medium | 86.80 | 83.50 | 82.40 | 90.70 | 85.60 | 86.40 | 88.80 |
| hun | small | 88.30 | 83.60 | 83.60 | 91.30 | 85.30 | 84.50 | 84.80 |
| hye | small | 87.00 | 90.60 | 92.00 | 95.60 | 94.30 | 91.90 | 88.70 |
| ita | small | 92.60 | 97.60 | 97.90 | 98.40 | 94.80 | 95.10 | 95.80 |
| jap | medium | 97.00 | 94.50 | 94.70 | 97.00 | 94.90 | 93.60 | 93.60 |
| kat | small | 85.80 | 85.20 | 85.80 | 88.80 | 83.20 | 80.90 | 84.90 |
| klr | medium | 100.00 | 99.60 | 99.40 | 100.00 | 99.40 | 99.30 | 100.00 |
| mkd | medium | 96.10 | 96.60 | 93.60 | 98.20 | 92.60 | 93.20 | 94.20 |
| nav | small | 63.70 | 65.30 | 58.30 | 72.40 | 56.40 | 53.60 | 59.40 |
| rus | small | 89.70 | 93.50 | 90.20 | 94.70 | 86.10 | 87.80 | 87.30 |
| san | small | 81.30 | 72.90 | 72.40 | 84.30 | 68.20 | 67.30 | 75.60 |
| sme | medium | 78.00 | 77.20 | 73.80 | 86.00 | 70.80 | 70.80 | 77.50 |
| spa | medium | 93.90 | 93.30 | 92.30 | 95.10 | 93.30 | 93.00 | 93.70 |
| sqi | medium | 95.20 | 90.80 | 90.00 | 97.50 | 92.90 | 89.40 | 82.90 |
| swa | medium | 96.40 | 97.40 | 93.10 | 97.70 | 97.20 | 97.70 | 91.40 |
| tur | small | 94.70 | 90.20 | 91.10 | 95.80 | 94.30 | 90.20 | 94.50 |
| **Average** | | 88.54 | 87.52 | 85.86 | 92.43 | 85.27 | 84.29 | 85.44 |

Table 11: **Oracle Accuracies (2023 data).** Accuracies of each method if an oracle were used to select among the hypotheses returned from beam search. In the case of BL10, an oracle chooses out of the 10 candidates returned from L2R *and* R2L's beam search; in the case of BL2, an oracle chooses between the best L2R and best R2L hypothesis.

| | Model Size | Baselines | | | Standalone | | | Reranker | | |
|---|---|---|---|---|---|---|---|---|---|---|
| | | L2R | R2L | BL2 | xH | xH-Rand | MML | xH | xH-Rand | MML |
| afb | small | 75.20 | 78.10 | 80.70 | 84.10 | 81.00 | 78.70 | **84.60** | 82.40 | 79.20 |
| amh | medium | 84.40 | **89.30** | 88.90 | 88.90 | 88.50 | 83.40 | 88.60 | 88.40 | 83.40 |
| arz | small | 87.20 | 88.10 | **89.20** | 89.10 | 87.80 | 87.50 | 88.70 | 87.50 | 87.40 |
| bel | large | 70.30 | 70.10 | **73.50** | 72.90 | 70.80 | 72.80 | 72.90 | 72.30 | 72.90 |
| dan | medium | 88.40 | 87.80 | **88.80** | 86.50 | 87.70 | 83.60 | 87.50 | 87.40 | 83.60 |
| deu | large | 73.10 | 78.00 | 79.70 | 80.20 | 80.90 | **81.10** | 79.70 | 80.50 | 81.00 |
| eng | medium | 94.50 | 94.00 | 95.60 | 95.70 | 95.80 | 95.80 | 95.70 | **95.90** | 95.80 |
| fin | medium | 78.30 | 74.00 | 79.20 | 83.60 | 85.10 | 77.70 | 82.70 | **85.90** | 77.90 |
| fra | small | 63.70 | 67.70 | 69.30 | 71.70 | 72.10 | 71.60 | **72.90** | 72.30 | 71.50 |
| grc | medium | 53.20 | 39.10 | 48.90 | **56.00** | 48.90 | 53.50 | **56.00** | 49.00 | 53.30 |
| heb | large | 91.14 | 87.41 | **92.95** | 92.45 | 89.12 | 84.09 | 92.45 | 89.22 | 84.09 |
| hebu | medium | 78.50 | 74.10 | 77.30 | 83.70 | 86.10 | 75.00 | 83.60 | **86.20** | 75.00 |
| hun | small | 77.70 | 66.10 | 76.30 | 84.30 | 83.80 | 79.30 | **85.00** | 84.30 | 80.10 |
| hye | small | 82.00 | 86.40 | 88.40 | 94.20 | 90.60 | 86.50 | **94.30** | 91.30 | 86.20 |
| ita | small | 89.30 | 93.90 | **95.80** | 94.40 | 94.30 | 92.70 | 93.70 | 94.70 | 92.70 |
| jap | medium | 93.80 | 91.00 | 92.80 | **94.90** | 92.70 | 92.30 | **94.90** | 93.60 | 92.10 |
| kat | small | 79.70 | 79.50 | **84.10** | 81.30 | 79.80 | 81.40 | 82.90 | 80.80 | 81.10 |
| klr | medium | **99.40** | 98.30 | **99.40** | **99.40** | 99.20 | **99.40** | **99.40** | 99.20 | **99.40** |
| mkd | medium | 89.70 | 92.00 | 91.90 | 92.10 | 92.80 | 91.40 | 92.40 | **93.20** | 91.50 |
| nav | small | 53.70 | 48.90 | 54.00 | 55.10 | 50.40 | **57.10** | 55.60 | 52.10 | 57.00 |
| rus | small | 82.10 | 84.90 | **87.40** | 84.20 | 85.30 | 82.10 | 85.50 | 86.80 | 83.30 |
| san | small | 61.50 | 60.70 | 63.30 | **67.70** | 65.50 | 59.60 | 65.90 | 66.80 | 60.60 |
| sme | medium | 63.40 | 65.30 | 69.90 | 67.40 | 67.30 | **75.60** | 67.30 | 67.30 | 75.20 |
| spa | medium | 90.30 | 91.20 | 90.90 | 93.20 | 93.00 | 93.40 | 93.30 | 93.00 | **93.50** |
| sqi | medium | 85.00 | 84.40 | 87.60 | 91.00 | 88.30 | 82.50 | **91.90** | 87.90 | 82.40 |
| swa | medium | 92.70 | 92.90 | 93.10 | 96.60 | 96.90 | 90.50 | 96.60 | **97.30** | 90.50 |
| tur | small | 88.80 | 87.40 | 90.90 | 94.00 | 89.50 | 89.90 | **94.20** | 89.30 | 89.90 |
| **Average** | | 80.26 | 79.65 | 82.59 | 84.25 | 83.08 | 81.43 | **84.38** | 83.50 | 81.50 |
| **Number ≥** | | | | | **19/27** | 14/27 | 9/27 | 18/27 | 15/27 | 9/27 |

Table 12: **Random Cross-Entropy Accuracies (2023 data).** A number is colored in green if it improves over BL2, and a number is **bold** if it is the best for the language.

| Language | Model Size Chosen | L2R Val. Accuracies | | | R2L Val. Accuracies | | | Bidirectional Val. Accuracies | | |
|---|---|---|---|---|---|---|---|---|---|---|
| | | S | M | L | S | M | L | xH | xH-Rand | MML |
| afb | small | 74.5 | 77.4 | 74.8 | 79.9 | 76.6 | 74.4 | 83.7 | 81.1 | 81.6 |
| amh | medium | 81.1 | 84.8 | 83.0 | 81.8 | 83.9 | 82.1 | 88.7 | 90.1 | 86.1 |
| arz | small | 87.9 | 89.3 | 88.3 | 89.0 | 87.4 | 87.9 | 89.6 | 89.4 | 88.2 |
| bel | large | 73.1 | 73.3 | 74.5 | 73.2 | 75.5 | 74.8 | 76.0 | 77.1 | 76.4 |
| dan | medium | 88.9 | 90.0 | 89.8 | 89.4 | 89.9 | 88.5 | 87.3 | 90.3 | 87.2 |
| deu | large | 79.3 | 80.1 | 78.6 | 76.7 | 76.9 | 79.6 | 80.5 | 81.1 | 77.8 |
| eng | medium | 94.8 | 95.9 | 94.8 | 94.9 | 94.6 | 93.8 | 95.2 | 95.9 | 95.0 |
| fin | medium | 93.7 | 96.8 | 92.4 | 92.6 | 90.8 | 91.2 | 98.1 | 96.3 | 94.8 |
| fra | small | 76.3 | 80.7 | 73.5 | 79.7 | 74.9 | 72.4 | 83.2 | 82.5 | 80.0 |
| grc | medium | 57.1 | 62.5 | 57.4 | 56.5 | 57.5 | 54.7 | 67.2 | 63.3 | 66.6 |
| hebu | medium | 91.3 | 92.7 | 90.7 | 92.8 | 92.4 | 91.6 | 94.3 | 93.9 | 93.5 |
| heb | large | 89.9 | 90.7 | 90.1 | 85.4 | 86.6 | 89.9 | 93.3 | 93.0 | 90.4 |
| hun | small | 87.3 | 85.9 | 81.7 | 77.3 | 77.3 | 74.7 | 88.7 | 89.1 | 79.5 |
| hye | small | 89.1 | 86.5 | 84.3 | 84.6 | 75.4 | 70.1 | 95.1 | 92.8 | 94.4 |
| ita | small | 95.8 | 94.7 | 94.2 | 94.0 | 85.4 | 88.2 | 97.3 | 96.3 | 96.8 |
| jap | medium | 89.6 | 89.8 | 88.6 | 88.6 | 89.0 | 90.2 | 92.9 | 92.6 | 92.0 |
| kat | small | 81.6 | 79.4 | 79.8 | 76.6 | 74.7 | 71.3 | 80.0 | 79.1 | 80.4 |
| klr | medium | 99.6 | 99.4 | 98.8 | 98.7 | 99.4 | 99.3 | 99.8 | 99.8 | 99.8 |
| mkd | medium | 93.2 | 94.0 | 92.6 | 91.3 | 93.6 | 90.6 | 96.0 | 95.7 | 93.9 |
| nav | small | 59.3 | 53.5 | 50.1 | 56.3 | 54.4 | 54.9 | 57.6 | 59.0 | 59.5 |
| rus | small | 88.3 | 87.5 | 86.3 | 87.9 | 87.3 | 85.7 | 88.6 | 87.5 | 86.5 |
| sme | medium | 72.2 | 74.7 | 72.0 | 70.2 | 70.3 | 62.5 | 76.0 | 79.0 | 81.4 |
| spa | medium | 95.2 | 96.0 | 91.3 | 94.5 | 94.0 | 94.1 | 98.5 | 97.5 | 95.9 |
| sqi | medium | 89.2 | 89.7 | 79.5 | 90.4 | 90.6 | 74.8 | 93.9 | 93.4 | 89.5 |
| swa | medium | 97.1 | 96.5 | 92.4 | 96.8 | 97.6 | 95.5 | 97.6 | 97.6 | 97.3 |
| tur | small | 92.1 | 92.1 | 87.5 | 90.7 | 82.4 | 70.7 | 97.1 | 94.6 | 90.0 |
| san | small | 67.5 | 63.2 | 57.6 | 63.1 | 46.2 | 47.9 | 76.2 | 75.6 | 71.4 |

Table 13: **Validation Accuracies (2023 data).** Validation accuracies for each language in the 2023 dataset. Validation accuracies on unidirectional models are used for hyperparameter selection. The bidirectional validation accuracies (xH, xH-Rand, MML) are reported for the chosen model size for each language.

| Language | Model Size Chosen | L2R Val. Accuracies | | | R2L Val. Accuracies | | | xH |
|---|---|---|---|---|---|---|---|---|
| | | S | M | L | S | M | L | |
| ang | large | 59.5 | 64.1 | 63.9 | 62.7 | 64.5 | 64.9 | 64.0 |
| ara | small | 75.0 | 74.5 | 75.3 | 75.9 | 75.7 | 74.9 | 74.7 |
| asm | medium | 83.6 | 84.4 | 85.0 | 76.9 | 88.0 | 85.1 | 85.6 |
| evn | small | 52.1 | 50.9 | 50.0 | 53.4 | 50.6 | 49.3 | 53.0 |
| got | large | 81.0 | 81.7 | 81.8 | 80.0 | 79.0 | 82.1 | 78.3 |
| heb | large | 26.1 | 27.7 | 27.6 | 31.2 | 31.3 | 31.7 | 28.2 |
| hun | medium | 67.1 | 75.0 | 73.4 | 70.6 | 75.9 | 75.0 | 76.5 |
| hye | medium | 91.2 | 93.4 | 92.1 | 91.0 | 93.9 | 93.4 | 91.7 |
| kat | large | 86.0 | 88.7 | 87.7 | 86.0 | 89.2 | 90.4 | 92.6 |
| kaz | medium | 65.6 | 67.7 | 67.3 | 54.0 | 61.2 | 60.3 | 57.7 |
| khk | medium | 38.0 | 39.8 | 39.2 | 39.2 | 39.7 | 39.2 | 39.2 |
| kor | large | 56.9 | 57.6 | 58.6 | 57.8 | 59.6 | 58.9 | 56.5 |
| krl | medium | 64.6 | 66.8 | 64.7 | 65.5 | 65.7 | 67.0 | 62.7 |
| lud | medium | 59.8 | 71.5 | 65.2 | 71.5 | 76.6 | 55.3 | 60.5 |
| non | large | 85.7 | 88.9 | 86.9 | 87.4 | 86.5 | 88.8 | 89.1 |
| pol | medium | 90.3 | 91.9 | 90.3 | 90.5 | 90.5 | 91.0 | 91.0 |
| poma | large | 49.4 | 52.7 | 53.6 | 49.3 | 55.4 | 57.5 | 52.9 |
| slk | large | 90.7 | 91.4 | 92.4 | 92.8 | 92.7 | 92.9 | 92.7 |
| tur | medium | 96.0 | 96.5 | 96.4 | 95.7 | 97.2 | 96.0 | 97.3 |
| vep | large | 63.0 | 61.6 | 63.5 | 59.7 | 59.6 | 60.9 | 64.1 |

Table 14: **Validation Accuracies (2022 data).** Validation accuracies for each language in the 2022 dataset. Validation accuracies on unidirectional models are used for hyperparameter selection. The xH validation accuracies are reported for the chosen model size for each language.