# OpenReview forum: "A Framework for Bidirectional Decoding: Case Study in Morphological Inflection"
_EMNLP/2023/Conference — EMNLP 2023 Findings_

### Official Review · Reviewer_qVTT · 2023-08-01

**Soundness:** 3

**Excitement:**

4: Strong: This paper deepens the understanding of some phenomenon or lowers the barriers to an existing research direction.

**Paper Topic And Main Contributions:**

This paper introduce a bidirectional decoding framework for morphological inflection. Different from previous works, this work generates sequences from both sides to the middle and utilize an efficient dynamic programming algorithm for training, combing the merits of both R2L and L2R decoders.
Experimental results show that the model performs particularly well on long sequences and achieve good results on SIGMORPHON inflection tasks.

**Questions For The Authors:**

Question A: could you elaborate on how the dynamic programming is achieved on the attention mechanism?

**Reasons To Accept:**

1. I think the application of bidirectional decoding to morphological inflection is new and reasonable.  The model is able to decide which generation order is best for each sequence, and can produce part of a sequence from each direction. So the search space is greatly expanded comparing with pure L2R decoding.

2. The experiment is well designed and related work comparison is sufficient, achieving SOTA on SIGMORPHON 2022 shared task.


**Reasons To Reject:**

1. I am a little confused by the application of dynamic programming. Since the complexity of attention mechanism in transformer is not polynomial, so it is not feasible for dynamic programming.  As the author mentioned at the end of Sec4, "since this architecture does have cross-attention ...  the decoder hidden states must be recomputed at each time step to allow for DP".  I think it would be very time consuming for this computation.  So I suggest the author to list out the decoding speed results comparison to show a full picture of this work.


**Reproducibility:**

3: Could reproduce the results with some difficulty. The settings of parameters are underspecified or subjectively determined; the training/evaluation data are not widely available.

**Reviewer Confidence:**

4: Quite sure. I tried to check the important points carefully. It's unlikely, though conceivable, that I missed something that should affect my ratings.

---

> ### Author Rebuttal · Authors · 2023-08-29
>
> **Dynamic programming**
>
> The reviewer writes “the complexity of attention mechanism in transformer is not polynomial” - this is incorrect: the standard transformer’s attention mechanism has $O(|y|^2)$ complexity, which is polynomial. For our bidirectional method, the DP table $F$ is of size $|y|\times |y|$, and each row represents a token in the L2R direction and each column represents a token in the R2L direction (the top left corner is empty prefix-empty suffix). DP will work as long as $F[i,j]$ depends only on a fixed number of previously computed cells and not on all possible paths to that cell (which would be exponential). In our case, $F[i,j]$ depends only on $F[i,j-1]$ and $F[i-1,j]$, and the local probabilities involved with predicting the $i$th L2R letter and the $j$th R2L letter. Crucially, these local probabilities cannot rely on previous hidden states, since these would be based on the path through the table up to this cell; instead, we recompute all hidden states at each cell $F[i,j]$, which incurs an additional $O(|y|)$ in complexity. Making sure that the local probabilities are path-independent is what enables DP to work on this task.
>
> **Inference Times**
>
> It turns out that the increase in runtime only has a minor impact on empirical decoding speed. As part of this rebuttal, we timed inference on the trained models for the baselines (L2R and R2L) and the bidirectional variants. The following table reports the average time it takes to decode 50 examples across all languages for each model type (on 4 NVIDIA V100 GPU’s). In the final version of our paper, we plan to present this information with box plots or a similar figure.
>
> |  Model  | Time per 50 examples (s) |
> | -------- | ------- |
> | L2R | 0.276   |
> | R2L | 0.277     |
> |xH|0.724|
> | xH-Rand    | 0.738   |
> |MML	|	0.772|
>
> Clearly, the need to recompute hidden states at each step slows inference down by about 3x. However, in practice we barely notice the difference on this task, as the test sets have only 1,000 examples. Given the strong outperformance of the bidirectional methods over the unidirectional baselines (and even over the naive bidirectional baseline BL2), one must therefore make a tradeoff between time and performance. Nonetheless, a major thrust for future work in this area is efficiency: how we can maintain the appealing aspects of the bidirectional decoding framework (e.g. decoding from the outside-in and the strong performance gains) while speeding up the model.
>
> **Note about SOTA on 2023 data**
>
> Finally, we would like to make the reviewer aware that the official shared task paper (Goldman et al., 2023) was released in the time since we submitted this paper. It turns out that a total of three teams submitted, and only our system beats the (neural) baseline, which had an average accuracy of 81.6 compared with ours of 84.3. The best submissions from the other teams were 76.9 (TUB) and 72.4 (AZ) respectively. We will include this comparison in the final report, since it shows that we set SOTA on the 2023 shared task (as well as the 2022 shared task, which is already included in our paper).

---

### Official Review · Reviewer_4BJv · 2023-08-04

**Soundness:** 3

**Excitement:**

3: Ambivalent: It has merits (e.g., it reports state-of-the-art results, the idea is nice), but there are key weaknesses (e.g., it describes incremental work), and it can significantly benefit from another round of revision. However, I won't object to accepting it if my co-reviewers champion it.

**Paper Topic And Main Contributions:**

The paper is a work on using bidirectional decoding to address unidirectional decoding issues. This paper proposes a bidirectional decoding method that chooses to generate a token on the left, generate a token on the right, or join the left and right sequences. The proposed method is tested on morphological inflection task.

**Reasons To Accept:**

The paper proposes a novel bidirectional decoding framework that chooses to generate a token on the left, generate a token on the right, or join the left and right sequences.

The proposed approach demonstrates significant improvements in accuracy compared to conventional left-to-right models, particularly in the context of morphological inflection tasks.

The paper sets a new state-of-the-art (SOTA) result for the 2022 shared task, showcasing the practical value and relevance of the proposed framework for real-world NLP applications.

**Reasons To Reject:**

Lack of comparison systems that use bidirectional approaches: In (Zhou et al., 2019b; Imamura and Sumita, 2020), the model decodes interactively from both directions at the same time and meets in the center. In contrast, this model decodes interactively from both directions and can meet at any position predicted by the model itself, with the ability to predict the decoding direction at each step. However, the comparison is made with just a naive bidirectional baseline that returns either the L2R or R2L hypothesis.

Lack of ablation studies: The proposed model can predict the decoding direction at each step, allowing it to decode interactively from both directions and now can meet at any position predicted by the model itself, rather than just meeting in the center (Zhou et al. 2019b). But how do we know empirical improvement comes from being able to predict the decoding direction and being able to meet at any position predicted by the model?

**Reproducibility:**

3: Could reproduce the results with some difficulty. The settings of parameters are underspecified or subjectively determined; the training/evaluation data are not widely available.

**Reviewer Confidence:**

3: Pretty sure, but there's a chance I missed something. Although I have a good feel for this area in general, I did not carefully check the paper's details, e.g., the math, experimental design, or novelty.

---

> ### Author Rebuttal · Authors · 2023-08-29
>
> **Comparison with other bidirectional approaches**
>
> The only iteration of the SIGMORPHON shared task where a team submitted a bidirectional approach was 2020, where one team (Canby et al., 2020) used the method of Zhou et al., 2019a. In our initial experiments, we did train xH models using our framework on three of the datasets from that year, and all three of them beat the approach from Canby et al., 2020: Old English (79.69 vs. 78.40), Turkmen (85.78 vs. 85.60), and Sanskrit (93.81 vs. 93.40). However, we ultimately decided to use more recent shared task datasets (2022 and 2023) for this paper, and so we could not directly compare with the results from 2020 in a systematic way.
>
> As the reviewer writes, we chose not to re-implement previous bidirectional decoders for this task. We chose instead to spend our energy training and fine-tuning high-quality L2R and R2L baselines. These are standard unidirectional transformers and actually beat previous SOTA on the 2022 datasets: the L2R baseline beats SOTA by about 1.9 points and the R2L baseline beats it by about 3.2 points, as shown in Table 2 of the paper. We also opt to focus our comparisons on SOTA rather than previous bidirectional decoders, although we do also provide a simple bidirectional baseline BL2 as a benchmark. In the end, even though it may be interesting to compare with other bidirectional decoders, we feel that it is sufficient to compare with well-trained baselines, BL2, and SOTA on the years in question.
>
> **Ablation Studies**
>
> The reviewer asks “how do we know empirical improvement comes from being able to predict the decoding direction and being able to meet at any position predicted by the model?” This is a very good question, and in our rebuttal we provide two simple insights into this as follows; we plan to include these in the final version of our paper.
>
> *Study 1*
>
> First, we simply compute the “confusion matrix” for our bidirectional model xH and our best baseline BL2 over all languages: this shows the number of examples that both models get correct, that one model is correct and the other is incorrect, and that both models get incorrect:
>
> | |xH✔︎ | xH✗ |
> | -------- | --------| ------- |
> |BL2✔︎|21210|1082|
> |BL2✗|1530|3171|
>
> The xH model clearly has a net improvement of 448 (1530 – 1082) examples over the BL2 baseline. The question is whether the 1,530 examples that the xH model gets right and BL2 gets wrong are due to xH’s ability to meet at any position in the output string. We thus compute the percentage of examples in each of these cells for which the xH model chooses to generate from both directions (rather than in a purely L2R or R2L direction, which is what BL2 does):
>
> |	|xH✔︎|	xH✗|
> | -------- | --------| ------- |
> |BL2✔︎	|76.5%|94.2%|
> |BL2✗|92.2%|87.6%|
>
> We see that in all cells, the xH model generally favors decoding words from both sides; however, it strongly favors this in the two diagonal cells, where one model is correct and the other is wrong. Most importantly, 92.2% of examples that are corrected by xH (bottom left cell) are decoded from both sides, compared with 76.5% of examples that both models get right. This suggests that the improvement over BL2 does indeed have to do with decoding order. As a side note, about 94.2% of examples that the xH model gets incorrect and BL2 gets correct (top right cell) are decoded from both sides; hence, in some cases decoding from both sides hurts accuracy. However, the confusion matrix shows that xH has a net improvement of 448, so, when looked at in its entirety, the new decoding method helps.
>
> *Study 2*
>
> Here, we perform a more traditional ablation study: we take our bidirectional models (xH and MML) and force decoding to be either fully L2R or fully R2L. We do this by setting the log probabilities of the opposite direction to $-\infty$ so that the beam search algorithm cannot choose them. The results are shown below in a format similar to Table 1 of the paper (averaged over all languages):
>
> | Model | Avg Acc. |
> | -------- | ------- |
> |L2R|80.26|
> |R2L|79.65|
> |BL2|82.59|
> |xH|84.25|
> |MML|81.43|
> |xH-forced L2R decoding|71.05|
> |xH-forced R2L decoding|77.31|
> |xH2|78.42|
> |MML-forced L2R decoding|4.68|
> |MML-forced R2L decoding|0.07|
> |MML2	|2.33|
>
> Here, xH2 and MML2 are analogous to BL2 in that they pick the best option from the L2R-decoded sequence and the R2L-decoded sequence.
>
> Clearly, the bidirectional models perform poorly when they are not allowed to be decoded as bidirectional models. This is particularly clear for MML, and it makes sense this would be detrimental for it: since it was trained with marginal likelihood, it can learn the order probabilities and thus assign low probability to some orderings (see Section 3.4 of the paper). Clearly MML does not favor unidirectional orders (also explored in Appendix G.3). Thus, the bidirectional nature of the MML model is crucial to its success.
>
> The xH model, on the other hand, does not suffer detrimental accuracy drops when forced to decode in the L2R or R2L direction; this is because it was trained to treat all orderings equally, so it should do reasonably well on any given ordering. However, the xH model with unidirectional decoding does suffer a drop between about 5 and 13 points compared with the xH model with full bidirectional decoding. This shows that following the generation procedure of Section 3.5 is critical to the success of the xH model, which is the model that sets SOTA.
>
> **Note about SOTA on 2023 data**
>
> Finally, we would like to make the reviewer aware that the official shared task paper (Goldman et al., 2023) was released in the time since we submitted this paper. It turns out that a total of three teams submitted, and only our system beats the (neural) baseline, which had an average accuracy of 81.6 compared with ours of 84.3. The best submissions from the other teams were 76.9 (TUB) and 72.4 (AZ) respectively. We will include this comparison in the final report, since it shows that we set SOTA on the 2023 shared task (as well as the 2022 shared task, which is already included in our paper).

---

### Official Review · Reviewer_HEuK · 2023-08-12

**Soundness:** 4

**Excitement:**

3: Ambivalent: It has merits (e.g., it reports state-of-the-art results, the idea is nice), but there are key weaknesses (e.g., it describes incremental work), and it can significantly benefit from another round of revision. However, I won't object to accepting it if my co-reviewers champion it.

**Paper Topic And Main Contributions:**

The authors have introduced a new approach for bidirectional decoding, where a model can decide the (L2R or R2L) order of generation for each subsequence. They've incorporated this into a transformer-based architecture, enabling efficient training through a dynamic programming algorithm. It seems to be able to implicitly learn the morpheme structure of words.

**Reasons To Accept:**

1. Improvement Over Baselines: The paper introduces a bidirectional model that consistently outperforms baselines, including unidirectional models and a baseline (BL2) that selects higher probabilities between L2R and R2L hypotheses.
2. Effective Reranking Strategies: The paper explores different reranking strategies based on marginal probabilities, both under the proposed xH model and the Maximum Marginal Likelihood (MML) approach. These strategies show enhancement in performance compared to the baselines, showcasing the potential of these techniques for improving bidirectional decoding.
3. Language and Output-Length Agnostic Improvement: The results show improvement across various languages and different output form lengths. This demonstrates the robustness of the proposed bidirectional approach across different linguistic contexts.
4. Decoding Strategies: The paper introduces effective decoding strategies, such as beam search, for optimizing joint probabilities. While direct likelihood-based decoding might not be computationally feasible, the paper adopts practical strategies for achieving accurate decoding.
6. Clear Presentation of Results: The paper presents a comprehensive analysis based on the results, and it uses many plots, making it easy for readers to follow the research methodology, results, and implications.

**Reasons To Reject:**

1. ~~Reproducibility: The paper does not provide information on code availability, making it difficult to assess the reproducibility of the results. It also doesn't provide any code or mention the possibility of future release of the code. Implementing the model and the some of the hyperparameter schedules (see their appendix) seems far from trivial, so it would be hard for the model to be adopted by the research community without some well-documented code.~~

2. The model is clearly useful for morphological inflection; it seems to be able to both inflect and be used for segmentation (despite only being trained for one of the tasks). However, it's suitability for other sequence generation tasks seems limited, and so I'm skeptical of the generality of the framework. I'm hard pressed to think of tasks outside of word-level ones where there is a natural split between two components like there is between stem and affix. As mentioned in the limitations section, this framework incurs significant computational burden, going from |y|^2 to |y|^4. Thus, the framework exacerbates rather than alleviates one of the most poignant critique of transformer-based models which is their quadratic time and space complexity, and is thus (in it's current form) unlikely to be adopted outside of word-level tasks like those regularly presented in SIGMORPHON shared task. Overall, I believe that the authors claim that this is a generally practical bidirectional framework is a strong overreach.

3. For all the complexity in implementing and training this model to convergence, the improvement over the naive bidirectional baseline (returning the best L2R or R2L hypothesis) is only modest (2.3 points).

**Reproducibility:**

4: Could mostly reproduce the results, but there may be some variation because of sample variance or minor variations in their interpretation of the protocol or method.

**Reviewer Confidence:**

3: Pretty sure, but there's a chance I missed something. Although I have a good feel for this area in general, I did not carefully check the paper's details, e.g., the math, experimental design, or novelty.

---

> ### Author Rebuttal · Authors · 2023-08-29
>
> **Reproducibility**
>
> We’d first like to point out a minor oversight of the reviewer regarding reproducibility: the reviewer writes “[The paper] doesn't provide any code or mention the possibility of future release of the code.” To the contrary, the footnote to our abstract on page 1 states “Our code will be made available on Github for the camera-ready version.” We commit to doing exactly this; we will publish the code as an extension to the fairseq library, so that interested researchers can easily use (and extend or modify) our method within a well-known framework on a variety of datasets.
>
> **Runtime**
>
> We do acknowledge the increase in training runtime (from $O(|y|^2$) to $O(|y|^4)$), which, as the reviewer says, could limit the applicability of the method to other tasks with longer sequences; we try to be up front about this point in the paper, both in the relevant section of the main paper (3.4) and in the Limitations section. Addressing this issue would be a key area of future work. We would however like to make a few comments on this topic:
> * In Appendix F, we actually provide an $O(|y|^3)$ method called xH-Rand that improves over the L2R baseline by an average of 3.24 points (compared with 4.12 for the $O(|y|^4)$ method). This shows that more efficient methods under the bidirectional framework are possible and still yield significant improvement over the baselines.
> * We designed our framework in such a way that it could be used to train an architecture with inner state preservation under the xH-Rand loss in $O(|y|^2)$ time, the same complexity as a standard transformer. Note that a model that uses inner state preservation *and* attention across the prefix and suffix would render DP impossible (see section 3.3); however, the probability factorization and generation procedure of decoding such a model from the outside-in would remain the same.
> * Such efficiency improvements are fertile ground for future research; this paper’s primary goal is to provide a starting point for such work and demonstrate the feasibility of a flexible bidirectional decoding framework.
>
> **Does the complexity of the model make the performance boost worth it?**
>
> The reviewer makes a practical point that training the more complicated bidirectional model xH-Rerank may not be worth the hassle compared with training BL2 (our simple bidirectional baseline), since its improvement is a “modest” 2.3 points. We would like to make the following points about this:
> * In the time since we submitted this paper, the official shared task paper (Goldman et al., 2023) was released. A total of three teams submitted, and only our system beats the (neural) baseline, which had an average accuracy of 81.6 compared with ours of 84.3. The best submissions from the other teams were 76.9 (TUB) and 72.4 (AZ) respectively. Hence, it seems that the magnitude of our system’s improvement is actually in line with the differences between other systems’ accuracies. We intend to include these results as a comparison in the final version of our paper.
> * As we show in the paper, our model sets SOTA on the 2022 shared task by 4.7 points. Therefore, our system is SOTA on both the 2022 and 2023 datasets. Further, our significance tests on the 2023 dataset show that xH-Rerank has a statistically significant degradation on only 2/27 languages, suggesting that our model’s improvements are real. Therefore, we believe that a practitioner who primarily cares about performance should favor our model over BL2.
> * We believe the larger question of whether it is worth training the full bidirectional model for this improvement over the simpler BL2 should be left for downstream researchers, whose use cases and goals may vary greatly.

---

### Meta-Review · Area_Chair_wPK3 · 2023-09-19

**Recommendation:** 5

**Metareview:**

The paper proposes a novel framework for decoding in the morphological inflection task. Instead of standard left-to-right decoding, the model chooses at each step whether to generate a token on the left, on the right, or merge the left and right sequences. The approach leads to improvements over baselines and SoTA models in the 2022 and 2023 shared tasks on morphological inflection. The reviewers agree that the model is novel, reasonable, scalable to other languages, and yields better results than (more traditional) left-to-right decoders. Some reviewers note that the paper lacks ablation studies, comparison to other bidirectional models, and the model complexity is higher, but the authors carefully, neatly, and in a very detailed way addressed all of them.
Overall, the paper provides a strong model for morphological inflection that might become a new SoTA. The paper should gain sufficient interest in the community of NLP/computational morphology.
Also, it's especially great to see that the paper authors care about time complexity and use dynamic programming!

---

### Decision · Program_Chairs · 2023-10-07

**Decision:**

Accept-Findings

**Comment:**

The paper proposes a novel framework for decoding in the morphological inflection task. Instead of standard left-to-right decoding, the model chooses at each step whether to generate a token on the left, on the right, or merge the left and right sequences. The approach leads to improvements over baselines and SoTA models in the 2022 and 2023 shared tasks on morphological inflection. The reviewers agree that the model is novel, reasonable, scalable to other languages, and yields better results than (more traditional) left-to-right decoders. Some reviewers note that the paper lacks ablation studies, comparison to other bidirectional models, and the model complexity is higher, but the authors carefully, neatly, and in a very detailed way addressed all of them.
Overall, the paper provides a strong model for morphological inflection that might become a new SoTA. The paper should gain sufficient interest in the community of NLP/computational morphology.
Also, it's especially great to see that the paper authors care about time complexity and use dynamic programming!